# Semisynthetic Sesquiterpene Lactones Generated by the Sensibility of Glaucolide B to Lewis and Brønsted–Lowry Acids and Bases: Cytotoxicity and Anti-Inflammatory Activities

**DOI:** 10.3390/molecules28031243

**Published:** 2023-01-27

**Authors:** Layzon A. Lemos da Silva, Louis P. Sandjo, Laura S. Assunção, Anne N. Prigol, Carolina D. de Siqueira, Tânia B. Creczynski-Pasa, Marcus T. Scotti, Luciana Scotti, Fabíola B. Filippin-Monteiro, Maique W. Biavatti

**Affiliations:** 1Post-Graduate Program of Pharmacy, Universidade Federal de Santa Catarina, Campus Universitário–Trindade, Florianópolis CEP 8840-970, SC, Brazil; 2Department of Chemistry, Universidade Federal de Santa Catarina, Campus Universitário–Trindade, Florianópolis CEP 88040-900, SC, Brazil; 3Department of Pharmaceutical Sciences, Universidade Federal de Santa Catarina, Campus Universitário–Trindade, Florianópolis CEP 88040-970, SC, Brazil; 4Post-Graduate Program in Natural and Synthetic Bioactive Products, Universidade Federal da Paraíba, Cidade Universitária–Castelo Branco III, João Pessoa CEP 58051-900, PB, Brazil; 5Department of Clinical Analysis, Universidade Federal de Santa Catarina, Campus Universitário–Trindade, Florianópolis CEP 88040-900, SC, Brazil

**Keywords:** sesquiterpene lactones, semisynthesis, anti-inflammatory activity, cytotoxic activity

## Abstract

Sesquiterpene lactone (SL) subtypes including hirsutinolide and cadinanolide have a controversial genesis. Metabolites of these classes are either described as natural products or as artifacts produced via the influence of solvents, chromatographic mobile phases, and adsorbents used in phytochemical studies. Based on this divergence, and to better understand the sensibility of these metabolites, different pH conditions were used to prepare semisynthetic SLs and evaluate the anti-inflammatory and antiproliferative activities. Therefore, glaucolide B (**1**) was treated with various Brønsted–Lowry and Lewis acids and bases—the same approach was applied to some of its derivatives—allowing us to obtain 14 semisynthetic SL derivatives, 10 of which are hereby reported for the first time. Hirsutinolide derivatives **7a** (CC_50_ = 5.0 µM; SI = 2.5) and **7b** (CC_50_ = 11.2 µM; SI = 2.5) and the germacranolide derivative **8a** (CC_50_ = 3.1 µM; SI = 3.0) revealed significant cytotoxic activity and selectivity against human melanoma SK-MEL-28 cells when compared with that against non-tumoral HUVEC cells. Additionally, compounds **7a** and **7c.1** showed strongly reduced interleukin-6 (IL-6) and nitrite (NOx) release in pre-treated M1 macrophages J774A.1 when stimulated with lipopolysaccharide. Despite the fact that hirsutinolide and cadinanolide SLs may be produced via plant metabolism, this study shows that acidic and alkaline extraction and solid-phase purification processes can promote their formation.

## 1. Introduction

Glaucolide (Figure 1) is a germacranolide subtype of sesquiterpene lactones (SLs) commonly found in the Asteraceae family whose structure contains a C-7/C-11 endocyclic α,β-unsaturated γ-lactone ring [1]. Hirsutinolide (Figure 1) and cadinanolide structural isomers—vernojalcanolide and vernomargolide (Figure 1)—are also found in Asteraceae, especially in the Vernonieae tribe [2,3,4]. The genesis of hirsutinolide and cadinanolide remains unclear in the scientific literature. While some phytochemical studies performed on the Vernonieae species claimed that these skeletons are natural metabolites [5,6,7], others reported that hirsutinolides and cadinanolides as presumable artifacts from the natural glaucolides produced during plant extraction and/or during the fractionation process [8,9,10]. Some previous studies carried out reactions mimicking the chromatographic purification of SLs, demonstrating that silica gel caused the conversion of glaucolide into hirsutinolide [11] and cadinanolide [12,13]. A similar goal was reached when various hirsutinolides and cadinanolides were obtained after treating glaucolide A in methanol (MeOH) or ethanol (EtOH) with alumina or silica gel [14]. Likewise, the acidic absorbent bentonite (composed of metallic oxides) was used to prepare 5β-hydroxy-hirsutinolide from glaucolide B [15].

Analysis of isolation procedures reported in the literature for glaucolides and hirsutinolides revealed that the crude extracts were obtained via alcoholic maceration and were purified through successive silica gel chromatographic columns [10,16,17]. Although in some cases hirsutinolides were obtained solely [18,19,20], other phytochemical studies led to the identification of hirsutinolides, cadinanolides, and derivatives bearing methyl and ethyl ether functionalities [21,22,23]. These results suggest that protic solvents such as MeOH and EtOH as well as acid and alkaline stationary phases seem to contribute to glaucolide conversion, producing artifacts via addition, substitution, and annulation reactions during the purification process [21,24].

SLs are well-known as cytotoxic and antiproliferative agents for cancer cells [25,26,27,28]. Some hirsutinolides inhibited the growth of glioblastoma (U251MG and SF259) and breast cancer (MDA-MB-231) human tumor cell lines by affecting the activation and function of STAT3 [29]. This transcriptional factor activates the malignant state and controls the expression of target genes crucial for cell proliferation, also playing a key role in different aspects of the tumorigenic process in several cancers [30,31]. Moreover, 5β-hydroxy-hirsutinolide induced apoptosis in MDA-MB-231, SK-LU-1 (lung cancer), and CaSki (cervical cancer, HPV 16 positive) cells by removing the anti-apoptotic Bcl-2 proteins from the cells and by increasing pro-apoptotic proteins Bax, active caspase-8, and active caspase-9 [15].

Among the different types of cancer, melanoma is considered the most aggressive skin cancer, and almost 290 thousand new cases were detected worldwide in 2018, causing about 60 thousand deaths [32]. The National Cancer Institute of Brazil [33] estimated about 1800 deaths in 2017 because of melanoma and suggested there will be almost 8500 new cases of melanoma (including 4200 men and 4250 women) per year between 2020 and 2022. SLs showed cytotoxic activity against different melanoma cell lines by inducing apoptosis cell death and affecting different pro-inflammatory, anti-apoptotic, and antioxidant intracellular mediators and signaling pathways [26,34,35]. Guaianolides, for example, were able to improve melanogenesis by increasing the melanin content and tyrosinase activity of B16 melanoma cells [36].

Based on the data presented above, this study was designed to explore the chemical transformation of glaucolide B in pH-dependent conditions using Brønsted–Lowry and Lewis acids and bases as catalysts. We herein report the preparation and structure elucidation of semisynthetic sesquiterpene lactones, as well as their in vitro anti-inflammatory potential and their antiproliferative activity in human melanoma SK-MEL-28 cancer cells.

## 2. Results and Discussion

### 2.1. Semisynthetic Modifications from Glaucolide B

The extraction process based on leaf washing with acetone was previously used to selectively obtain glaucolide B (**1**) from *Lepidaploa chamissonis* (Less.) H. Rob [37]. The obtained crude extract was purified using single-step centrifugal partition chromatography employing the M HEMWat (*n*-hexane/ethyl acetate/methanol/water, 5:6:5:6) biphasic solvent system [37]. The choice of this silica-gel-free purification technique aimed to avoid any degradation that could change the SL profile of *L. chamissonis*. As mentioned before, the silica gel stationary phase increases the possibility of structure modification of SLs during the chromatographic purification [14]. Glaucolide B (**1**) was subjected to various chemical modifications using different Lewis and Brønsted–Lowry acids (BiCl_3_, SOCl_2_, Ac_2_O, and TFA; pKa −0.25) and bases (DMAP, pKa 9.5; K_2_CO_3_, pKa 10.3) to explore its chemical behavior. These catalysts were selected based on their different reactivity as it depends on their solubility in organic solvents, acidity, and basicity. Their reactivity also depends on potential acidic hydrogen atoms in different functionalities.

Figure 2 shows glaucolide B (**1**), first treated with *N,N*-dimethylaminopyridine (DMAP) in MeOH under reflux conditions, producing two known decaline products, identified as 13-*O*-methylvernojalcanolide-8-*O*-acetate [38] (**2**) and 1,4-dihydroxy-5,8,10,13-tetraacetoxycadin-7(11)-en-6,12-olide [8] (**3**), with yields of 4.9% and 2.6%, respectively. As shown in the proposed mechanism (Figure 3), DMAP removed the γ-H of the butyrolactone ring, and the remaining pair of electrons rearranged, causing the elimination of the acetate ion at C-13 and the formation of an α-exomethylene γ-lactone derivative (**1.2**). The two nucleophilic species, MeOH and AcOH, underwent a 1,4-addition to the exomethylene of the derivative **1.4**, which rearranged, producing the vernojalcanolide derivatives **2** and **3**, respectively (Figure 3). The same condition was repeated using dichloromethane (CH_2_Cl_2_, aprotic solvent) to reduce the chance of a side reaction with MeOH, and then, compound **3** was obtained with a yield of 30.3%. Similar vernojalcanolides bearing a methacryloyloxy group at C-8 instead of an acetoxy were previously obtained when glaucolide A was treated with silica gel in MeOH or EtOH [12,14].

Moreover, glaucolide B (**1**) transformation was explored under Lewis acidic conditions. From this reaction, three new acetalic vernomargolides were obtained and identified as 5,8-diacetoxy-2-epi-vernomargolide-1,4-cyclosemiacetal (**4**), 5,8,10-triacetoxy-2-epi-vernomargolide-1,4-cyclosemiacetal (**5**), and 5,8-diacetoxy-2-epi-vernomargolide-1,4-cycloacetal-1,10-acetonide (**6**). The formation of these compounds was possible by treating glaucolide B (**1**) with bismuth chloride (BiCl_3_) and acetic anhydride (Ac_2_O) in CH_2_Cl_2_ (Figure 2). The chemical profile of the obtained products indicated that the mixture of BiCl_3_ and Ac_2_O causes the formation of sub-products including a bismuth acetoxychloride [Bi(OAc)Cl_2_] species and acetone, as shown in Figure 4A, involved in the production of the acetalic vernomargolides **4**, **5,** and **6**. In an earlier study, BiCl_3_ promoted the *C*-acetylation of trimethylsilyl enol ethers via acetyl chloride yielding β-diketones [39]. For vernomargolide derivatives, the epoxide at C-4/C-5 in glaucolide B (**1**) was opened via the nucleophilic addition at C-5 of the acetoxy [40] after complexation with Bi(OAc)Cl_2_, with the subsequent nucleophilic addition of the ketone to produce the anionic intermediate **1.5**. The resulting alkoxide of **1.5** performed a rearrangement via 1,3-hydrogen displacement, followed by annulation and allylic shift with Bi(OAc)Cl_2_ elimination (Figure 4B), producing compound **4**. Compound **5** was produced by eliminating bismuth hydroxy chloride [Bi(OH)Cl_2_] after the hydrolysis of the bismuth complex of the intermediate **4.2**—formed via complexation of the Bi(OAc)Cl_2_ with the C-10 tertiary ester in compound **4**—with the elimination of Ac_2_O (Figure 4B). Products structurally related to **4** and **5** (presenting a methacryloyloxy group at C-8) were previously prepared from glaucolide A by using boron trifluoride [41]. Additionally, an unexpected acetalic product (compound **6**) was obtained, although acetone was used neither for the reaction nor as a mobile phase for chromatographic purification. Therefore, acetone might be formed via sequential reactions including hydrolysis, *C*-acetylation, and decarboxylation from the mixture of BiCl_3_ and Ac_2_O (Figure 4A). Such reactions of hydrolysis, decarboxylation, and acetalization promoted by BiCl_3_ on carbonyl compounds were previously reported in [42,43,44]. Acetone was reacted via 1,2-addition with vicinal OH groups in **4.2** to produce a 2,2-dimethyl-1,3-dioxolane ring (compound **6**).

The importance of acetic anhydride for the conversion of glaucolide B (**1**) into cadinanolides was further explored by stirring **1** only with BiCl_3_ in CH_2_Cl_2_. Instead of a cadinanolide, glaucolide B (**1**) was converted into a known hirsutinolide analog named 5-hydroxy-hirsutinolide [15] (**7**) with a yield of 77.4%. Its formation might involve the epoxy ring opening and cyclization (Figure 2 and Figure 5) caused by bismuth(III) salt, as previously reported in [44,45]. The same hirsutinolide (**7**) was obtained with a lower yield (37.5%) when glaucolide B (**1**) was stirred together with trifluoroacetic acid (TFA) (Figure 2). While the alkaline conditions (use of DMAP) and the use of BiCl_3_/Ac_2_O caused the conversion of glaucolide into decaline scaffolds, reactions using Lewis and Brønsted–Lowry acids solely favored a rearrangement of glaucolide into the hirsutinolide analog.

Calculation of the electronic energies revealed that compound **7** was formed from glaucolide B (**1**) with a difference of enthalpy of −77.3527 kcal/mol (Figure 6). The energy proximity of both compounds suggests why phytochemical adsorbents such as silica gel, Florisil, and alumina convert glaucolide into hirsutinolide derivatives during the separation process. The energetic barrier for the chemical conversion suggests that the stability of a glaucolide-type might depend on the pH of the plant crude extract in the solvent. The preparation condition of this conversion indicates that hirsutinolide derivatives might be formed kinetically from glaucolides.

The semisynthetic product, compound **7,** was further submitted to chemical modifications to investigate the influence of alkaline and acidic conditions (Figure 7). Thus, hirsutinolide analog **7** treated in alkaline conditions (Figure 8) using potassium carbonate (K_2_CO_3_) produced a new spiro(tetrahydrofuran-γ-butyrolactone) derivative identified as 1,8-diacetoxy-1(4),7(10)-diepoxy-5-hydroxygermacr-11(13)-en-6(12)-olide (**8**, 33.7%). The formation of **8** seems to be promoted by the deprotonation of the hemiacetalic hydroxyl moiety at C-1 due to the alkaline pH (Figure 8A). The alkoxide intermediate (**7.1**), after transesterification of the C-10 acetoxy, rearranged via annulation with a nucleophilic addition at C-7 and allylic shift, eliminated the C-13 acetoxy group as potassium acetate (CH_3_COOK) and converted C-7 to a spiro carbon, producing an exocyclic methylene at C-11/C-13 (Figure 8B). Similar to derivative **8,** SLs (bearing a π-bond at C5/C-6 instead of a hydroxymethyne at C-5) were produced from glaucolide and hirsutinolide by using K_2_CO_3_ under reflux in dioxane [38,46]. In addition, a related compound was also obtained via the silica gel chromatographic purification of a *Vernonia chamaedrys* crude extract [46].

Compound **7** also furnished a new vernomargolide analog, identified as 8-acetoxy-vernomargolide (**9**, 67.2%), when treated with the Lewis acid BiCl_3_ under reflux in CH_2_Cl_2_ (Figure 7). As proposed in Figure 9, the mechanistic pathway for the formation of compound **9** included a rearrangement from hirsutinolide **7** to produce the tautomer ketone glaucolide intermediate **7.2**, under reflux. After BiCl_3_ complexation and the H-α elimination under the Lewis acid’s influence, transannular cyclization occurred via the attack of C-α on the C-β of the lactone ring to furnish the vernomargolide intermediate (**7.3**). Regioselective hydrolysis of the bismuth complex at C-10 occurred via the intermediate **7.3** to produce compound **9** after eliminating the ketone α-acetoxy moiety. To the best of our knowledge, this is the first time that vernomargolide **9** was produced from a hirsutinolide, although being previously described in the silica gel chromatographic purification of *Vernonia marginata* in [5].

Compounds **7–9** were subjected to acetylation using Ac_2_O with triethylamine (Et_3_N) and DMAP in CH_2_Cl_2_ (Figure 7) via the alcohol acylation mechanism [47,48], producing three new acetylated analogs, characterized as 5-acetoxy-hirsutinolide (**7a**, 37.8%), 1,5,8-triacetoxy-1(4),7(10)-diepoxy-germacr-11(13)-en-6(12)-o1ide (**8a**, 18.8%), and 5,8-diacetoxy-vernomargolide (**9a**, 65.7%), respectively. The acetylation occurred regioselectively in vernomargolide **9a** since the only secondary OH group in compound **9** was esterified. The two tertiary OHs at C-4 and C-10 remained free as in the starting material (**9**), presumably because of the neighboring steric effect promoted by the geminal methyl groups [49].

Compound **7** was also treated with thionyl chloride (SOCl_2_) in pyridine to substitute OH functions (Figure 7). No chlorinated products were found, but a known product of elimination was obtained with a 40.1% yield, which was characterized as 8,10,13-triacetoxy-1(4)-epoxy-germacra-1(2),5(6),7(11)-trien-6(12)-olide (**7b**). The olefin functions observed in **7b** indicated the occurrence of double dehydrohalogenation [50] via chlorination at C-1 and C-5 of 5-hydroxy-hirsutinolide (**7**). Compound **7b** was previously reported in the literature and was produced from a hirsutinolide-type SL [51,52].

Moreover, after treating hirsutinolide **7** with pyridinium chlorochromate (PCC) fixed on silica gel (1:1, *w*/*w*), instead of a mono-oxidized product at C-5, a new product with two oxidized carbons bearing a 5-oxo-6-hydroxy-γ-lactone was obtained (Figure 7). Diagnostic spectroscopic data led to the structure of 8,10,13-triacetoxy-1(4)-epoxy-6-hydroxy-5-oxo-germacr-7(11)-en-6(12)-olide (**7c**). It seems that, besides the oxidation of the hydroxyl at C-5, the lactone ring opening in the presence of silica gel (**7.5**) formed the OH group attached to γ-C (**7.8**), which was oxidized also by PCC to yield a 5,6-diketone derivative (**7.10**). This diketone intermediate **7.10** further reacted with the carboxylic acid to produce the 5-oxo-6-hydroxy-γ-lactone derivative (**7c**), as proposed in the mechanism illustrated in Figure 10.

As illustrated in Figure 7 and Figure 11, the oxidized hirsutinolide analog (**7c**) was further treated with BiCl_3_ in reflux, aiming to prepare a vernomargolide derivative. Unexpectedly, a new 1,5-oxo-4(7)-epoxy-6-hydroxy-γ-butyrolactone derivative bearing a macrocyclic hydrocarbon ring was obtained (13.2%). Its structure was assigned as 8,10-diacetoxy-4(7)-epoxy-6-hydroxy-1,5-oxo-germacr-11(13)-en-6(12)-olide (**7c.1**) based on its spectrometric data. The proposed mechanism pathway to form **7c.1** (Figure 11) involved a tandem reaction process including hydrolysis of the C-1 hemiacetal ring after bismuth complexation under reflux and a nucleophilic attack at C-7, causing the allylic elimination of Bi(OAc)Cl_2_. Although an oxidized decaline core was aimed to be the main product based on the preparation of compound **9** from the hirsutinolide **7**, the geometry of the oxidized carbons in the **7c** hydrocarbon macrocycle might disfavor the proximity between the C-2 and the α,β-unsaturation of the butyrolactone for the expected *C*-*C* bond formation, especially when compared with the presumably more favorable C-4/C-7 epoxy bridge rearrangement.

The structures of the prepared SLs were elucidated based on the spectroscopic data, including 1D and 2D nuclear magnetic resonance (NMR) and high-resolution mass spectrometry (HRESIMS) analyses (see the Experimental Section and Appendix A for more details). The spectroscopic data of known compounds were also compared with those reported in the literature.

### 2.2. Biological Activity

Glaucolide B (**1**) and its prepared derivatives were first evaluated for cytotoxic activities against three tumor cell lines including human melanoma (SK-MEL-28), human large-cell lung carcinoma (NCI-H460), and human glioblastoma cancer (SF295) cells, comparing with non-tumoral human umbilical vein endothelial cells (HUVECs) to determine the selectivity index (SI). As depicted in Table 1, hirsutinolides **7a** and **7b**, as well as germacranolide **8a**, showed significant cytotoxic activity and selectivity against human melanoma SK-MEL-28 cells, with CC_50_ values of 5.0 µM (SI = 2.5), 11.2 µM (SI = 2.5), and 3.1 µM (SI = 3.0), respectively. In addition, none of the evaluated compounds were active against cell lines NCI-H460 and SF295. Sesquiterpene lactones represent an important natural core for medicinal chemistry. The lactone ring and the α-methylene of SLs are two chemical features that can undergo 1,2- and 1,4-additions (Michael additions) to thiol-containing proteins [53], conferring on SLs numerous biological activities including antitumor and anti-inflammatory activities. SLs bearing this α-methylene-γ-lactone moiety exert antitumor effects by modifying the redox balance in cells and/or by inhibiting the signaling pathway of nuclear factor kappa B (NF-κB) [53]. NF-kB is known to be an anti-apoptotic metastasis promoter and is involved in the immune system response, mainly responsible for orchestrating the production of inflammatory cytokines and cell proliferation [54]. Interestingly, parthenolide, a germacranolide-type SL, has been reported to trigger apoptosis also in human melanoma SK-MEL-28 cells by activating a caspase-independent mechanism mediated by the nuclear translocation of apoptosis inducing factor, which is associated with several events such as reactive oxygen species generation, the depletion of protein thiols and glutathione, and the dissipation of the mitochondrial membrane potential [55]. Likewise, the SLs tomentosin and inuviscolide presented antiproliferative activity against human melanoma SK-MEL-28 cells by inducing G_2_/M arrest and apoptosis, whose mechanism was associated with a decrease in the mitochondrial membrane potential, caspase-3 activation, upregulations of p53 and p21 and down-regulation of the p65 subunit of NF-κB, and surviving [35].

Although compounds **7a** and **8a** are not analogous, the increase in lipophilicity via acetylation (Figure 7) seems to have a beneficial effect on cytotoxic activity. These findings suggest a positive influence on the cytotoxic activity against SK-MEL-28 cells provided by the incorporation of an acetoxy substituent in the C-5 position of both compounds in comparison with the respective starting materials containing a hydroxyl function in the same position. The hirsutinolide skeleton and its polyoxygenated functionalities provides these products with biological potentials more than their respective analog starting materials. Moreover, the change in physical properties such as lipophilicity due to new acetoxy groups in compounds **7a** and **8a** appears to improve their cell absorption, subsequently leading to the observed biological effect [56,57,58,59]. Compound **7b** was also more lipophilic than its starting material (**7**), and, interestingly, it also showed cytotoxicity even though it was two-fold less than **7a**.

To better understand the influence of increased lipophilicity, three other SLs previously purified from Vernonieae species [60], namely, piptocarphin A (**10**), glaucolide A (**11**), and diacetylpiptocarphol (**12**) (Figure 12), were also evaluated for cytotoxicity properties against the same tumoral cell lines (Table 1). Although presenting a hydroxyl group at C-10 besides the lipophilic olefin function at C-5/C-6, when compared with hirsutinolide **7**, its congeners (**10** and **12**) were also promisingly active (CC_50_ = 6.7 µM; SI = 1.9 and CC_50_ = 3.8 µM; SI = 3.3, respectively) against SK-MEL-28 cells.

The assessment of the anti-inflammatory activity of SL derivatives revealed that all compounds were anti-inflammatory to some degree (Figure 13). Pro-inflammatory M1-phenotype macrophages were used, via stimulation with lipopolysaccharide (LPS) through toll-like receptor 4 (TLR4), leading to the activation of the canonical NF-κB pathway [61]. Such triggering leads to the activation of MyD88-dependent TLR signaling, which is crucial for the expression of NF-κB, the key transcription factor of M1-macrophages and required for the induction of a large number of inflammatory genes, including those encoding IL-6 and NO [62]. Compounds **7a** and **7c.1** displayed a strong capacity to prevent NOx and IL-6 release from pre-treated M1-macrophages compared with control M1 cells (LPS + group) (approximately a 77% reduction, as shown in Figure 13A,B). Moreover, these compounds not only showed anti-inflammatory properties but also presented less cytotoxicity against macrophages (Appendix A)_._ A similar profile is registered in previous reports [16,22,63] for the well-known compounds (**10** and **11**). Even though antitumor and anti-inflammatory signaling may interact with each other, herein, we observed the potential for all compounds to disturb NO and IL-6 production and release, affecting not only the exacerbated inflammatory response but also cell proliferation [64].

## 3. Materials and Methods

### 3.1. General Experimental Procedures

UV spectra were recorded with a PDA detector from a Waters *Acquity UPLC H-class* system (Waters Co., Milford, MS, USA), and data were processed with *MassLynx 4.1* software. One-dimensional and two-dimensional NMR spectra were recorded with a Bruker *Fourier 300* spectrometer (Bruker Co., Billerica, MS, USA) using CDCl_3_ (Cambridge Isotope Laboratories, Inc., Andover, MS, USA) as a solvent, and data were processed with *TopSpin 3.2* software. NMR chemical shifts were referenced to tetramethylsilane (TMS). LC-HRESIMS spectra were measured with a Waters *Xevo G2-S QToF* mass spectrometer (Waters Co., Manchester, UK) equipped with a Zpray^TM^ electrospray ionization probe and a quadrupole-time of flight (QTof) analyzer, coupled to a Waters *Acquity UPLC H-class* system, using a Thermo Scientific^®^
*Hypersil GOLD* (100 × 3 mm, 1.9 µm) C_18_ column and a 0.5 mL/min elution of a gradient system combining 0.1% aqueous formic acid and acetonitrile. Acetonitrile HPLC grade (Tedia^®^ Brazil, Rio de Janeiro, Brazil), formic acid 85% *p.a.* grade (Vetec^®^ Química Fina, Rio de Janeiro, Brazil), and ultrapure reverse osmosis Milli-Q water (prepared using a Millipore^®^ Simplicity UV System from Millipore, France) were used for the eluent in the LC-MS analyses, while data were processed with *MassLynx 4.1* software. Column chromatography (CC) separations and solid-phase extraction (SPE) were performed using silica gel 60 (240−400 *mesh* from SiliCycle^®^, Quebec, QU, Canada) as support, and *p.a.* grade solvents, including *n*-hexane, dichloromethane (CH_2_Cl_2_), chloroform (CHCl_3_), and acetone (Vetec^®^ Química Fina, Rio de Janeiro, RJ, Brazil). Analytical and preparative TLC was carried out on silica gel 60 F_254_ precoated aluminum plates (SiliCycle^®^). Spots were observed under UV (254 and 366 nm) light and reacted with anisaldehyde-sulfuric acid [65] for analytical TLC, while for preparative TLC, spots were monitored under UV light, collected, and filtered through G3 sintered glass with acetone.

### 3.2. Semisynthetic Derivatives Production and Isolation

Glaucolide B (**1**), a pale yellow amorphous solid used as a starting material for SL semisynthetic derivatives production, was previously obtained from a *Lepidaploa chamissonis* (Less.) H. Rob. extract after centrifugal partition chromatographic isolation [37]. To investigate glaucolide B transformation as well as its derivatives under the influence of different acidic and basic conditions, catalysts with varied acid–base properties were used in transannular via allylic rearrangement cyclization reactions, in addition to oxidation, dehydrohalogenation, acetylation, and elimination reactions.

For the semisynthesis of 13-*O*-methylvernojalcanolide-8-*O*-acetate (**2**) and 1,4-dihydroxy-5,8,10,13-tetraacetoxycadin-7(11)-en-6,12-olide (**3**), condition *a(i)*, DMAP (1.5 mg, 0.01 mmol, 0.2 molar eq.) was added to a solution of glaucolide B (**1**; 30.1 mg, 0.07 mmol, 1 molar eq.) in MeOH (1 mL), and the mixture was refluxed at 60 °C for 3 h. TLC was used to monitor the reaction, which was stopped after the complete consumption of glaucolide B (**1**). The reaction mixture was concentrated under reduced pressure. The crude material was purified via silica gel CC using a gradient eluent system composed of acetone in *n*-hexane (20–50%, *v*/*v*) to obtain semisynthetic derivatives **2** (1.6 mg, 4.9%) and **3** (1.1 mg, 2.6%). Compound **2** was a yellowish amorphous solid; LC-UV [acetonitrile in 0.1% aqueous formic acid] λ_max_ 234 nm; ^1^H NMR(CDCl_3_, 300 MHz, at 295 K): δ 5.93 (1H, s, H-5), 5.87 (1H, dd, J = 4.0, 2.4 Hz, H-8), 4.42 (1H, d, J = 12.1 Hz, H-13a), 4.23 (1H, d, J = 12.1 Hz, H-13b), 3.35 (3H, s, OCH_3_-13), 3.32 (1H, dd, J = 16.0, 2.4 Hz, H-9a), 2.42–2.33 (1H, m, H-2a), 2.37–2.26 (1H, m, H-3a), 2.22 (3H, s, OCOCH_3_-10), 2.04 (1H, dd, J = 16.0, 4.0 Hz, H-9b), 2.04 (3H, s, OCOCH_3_-8), 1.98 (3H, s, OCOCH_3_-5), 1.92–1.81 (1H, m, H-3b), 1.73–1.66 (1H, m, H-2b), 1.70 (3H, s, CH_3_-14), and 1.41 (3H, s, CH_3_-15); ^13^C NMR (CDCl_3_, 75 MHz, at 295 K): δ 171.4 (CO, C-12), 171.4 (CO, OCOCH_3_-5), 170.5 (CO, OCOCH_3_-8), 168.9 (CO, OCOCH_3_-10), 157.2 (C, C-7), 129.5 (C, C-11), 88.9 (C, C-6), 84.2 (C, C-10), 76.8 (C, C-1), 75.9 (CH, C-5), 73.2 (C, C-4), 65.0 (CH, C-8), 63.1 (CH_2_, C-13), 58.7 (CH_3_, OCH_3_-13), 36.0 (CH_2_, C-3), 34.0 (CH_2_, C-9), 30.9 (CH_2_, C-2), 23.3 (CH_3_, C-15), 23.0 (CH_3_, OCOCH_3_-10), 21.2 (CH_3_, OCOCH_3_-8), 20.5 (CH_3_, OCOCH_3_-5), and 19.8 (CH_3_, C-14); HRESIMS: *m*/*z* 471.1859 [M+H]^+^ (calculated for C_22_H_31_O_11_, 471.1866, error_ppm_ = −1.5).

Compound **3** was a pale yellow amorphous solid; LC-UV [acetonitrile in 0.1% aqueous formic acid] λ_max_ 225 nm; ^1^H NMR (CDCl_3_, 300 MHz, at 295 K): *δ* 5.92 (1H, s, H-5), 5.86 (1H, dd, *J* = 4.1, 2.4 Hz, H-8), 5.12 (1H, d, *J* = 12.9 Hz, H-13a), 4.79 (1H, d, *J* = 12.9 Hz, H-13b), 3.32 (1H, dd, *J* = 16.1, 2.4 Hz, H-9a), 2.42–2.30 (1H, m, H-2a), 2.39–2.25 (1H, m, H-3a), 2.22 (3H, s, OCOCH_3_-10), 2.06 (1H, dd, *J* = 16.1, 4.1 Hz, H-9b), 2.05 (3H, s, OCOCH_3_-13), 2.03 (3H, s, OCOCH_3_-8), 1.98 (3H, s, OCOCH_3_-5), 1.90–1.77 (1H, m, H-3b), 1.73–1.60 (1H, m, H-2b), 1.70 (3H, s, CH_3_-14), and 1.41 (3H, s, CH_3_-15); ^13^C NMR (CDCl_3_, 75 MHz, at 295 K): *δ* 171.4 (*C*O, OCOCH_3_-5), 170.7 (*C*O, OCOCH_3_-8), 170.5 (*C*O, OCOCH_3_-13), 170.4 (*C*O, C-12), 168.9 (*C*O, OCOCH_3_-10), 157.4 (*C*, C-7), 128.0 (*C*, C-11), 89.4 (*C*, C-6), 84.2 (*C*, C-10), 76.8 (*C*, C-1), 75.9 (*C*H, C-5), 73.3 (*C*, C-4), 65.3 (*C*H, C-8), 55.0 (*C*H_2_, C-13), 35.8 (*C*H_2_, C-3), 33.8 (*C*H_2_, C-9), 30.7 (*C*H_2_, C-2), 23.3 (*C*H_3_, C-15), 22.9 (*C*H_3_, OCOCH_3_-10), 21.2 (*C*H_3_, OCOCH_3_-8), 20.7 (*C*H_3_, OCOCH_3_-13), 20.4 (*C*H_3_, OCOCH_3_-5), and 19.7 (*C*H_3_, C-14); HRESIMS: *m*/*z* 499.1794 [M+H]^+^ (calculated for C_23_H_31_O_12_, 499.1816, error_ppm_ = −4.4).

For reaction condition *a(ii)*, DMAP (1.8 mg, 0.01 mmol, 0.2 molar eq.) was added to a solution of glaucolide B (**1**; 30.3 mg, 0.07 mmol, 1 molar eq.) in anhydrous CH_2_Cl_2_ (1 mL). The evolution of this reaction was monitored with TLC, and the mixture was stirred for 30 min at room temperature. The reaction mixture was concentrated under reduced pressure and purified via silica gel CC using a gradient eluent system composed of acetone in *n*-hexane (20–100%, *v*/*v*), allowing the exclusive isolation of the semisynthetic derivative **3** (10.5 mg, 30.3%).

For the semisynthesis of 5,8-diacetoxy-2-epi-vernomargolide-1,4-cyclosemiacetal (**4**), 5,8,10-triacetoxy-2-epi-vernomargolide-1,4-cyclosemiacetal (**5**), and 5,8-diacetoxy-2-epi-vernomargolide-1,4-cycloacetal-1,10-acetonide (**6**), Ac_2_O (108 µL, 1.14 mmol, 10.0 molar eq.) and BiCl_3_ (161.8 mg, 0.51 mmol, 4.5 molar eq.) were added to a solution of glaucolide B (**1**; 50.0 mg, 0.11 mmol, 1 molar eq.) in anhydrous CH_2_Cl_2_ (1 mL) under an atmosphere of argon (Ar), and the mixture monitored with TLC was stirred for 2 h at room temperature (condition b). The reaction mixture was extracted with CH_2_Cl_2_ (3 × 15 mL) via liquid–liquid partition, and the organic layer was dried with anhydrous sodium sulfate (Na_2_SO_4_), filtered through filter paper, and concentrated under reduced pressure. The crude material was purified with silica gel CC using a gradient eluent system composed of acetone in CHCl_3_ (10–30%, *v*/*v*) to produce the isolation of the semisynthetic derivatives **4** (11.1 mg, 24.6%), **5** (13.9 mg, 27.8%), and **6** (3.4 mg, 6.8%).

Compound **4** was a pale yellow amorphous solid; LC-UV [acetonitrile in 0.1% aqueous formic acid] λ_max_ 214 nm; ^1^H NMR (CDCl_3_, 300 MHz, at 295 K): *δ* 6.47 (1H, s, H-13a), 5.68 (1H, s, H-13b), 4.88 (1H, d, *J* = 5.4 Hz, H-5), 4.87 (1H, dd, *J* = 3.2, 2.8 Hz, H-8), 4.38 (1H, d, *J* = 5.4 Hz, H-6), 3.20 (1H, brs, OH-1), 2.80 (1H, d, *J* = 4.2 Hz, H-2), 2.59 (1H, dd, *J* = 12.6, 4.2 Hz, H-3a), 2.225 (1H, d, *J* = 3.2 Hz, H-9a), 2.200 (1H, d, *J* = 2.8 Hz, H-9b), 2.17 (3H, s, OCOCH_3_-5), 2.07 (3H, s, OCOCH_3_-8), 1.65 (1H, d, *J* = 12.6 Hz, H-3b), 1.60 (1H, brs, OH-10), 1.33 (3H, s, CH_3_-14), and 1.31 (3H, s, CH_3_-15); ^13^C NMR (CDCl_3_, 75 MHz, at 295 K): *δ* 169.7 (*C*O, OCOCH_3_-5), 168.6 (*C*O, OCOCH_3_-8), 167.6 (*C*O, C-12), 136.8 (*C*, C-11), 125.5 (*C*H_2_, C-13), 106.1 (*C*, C-1), 83.4 (*C*H, C-6), 83.2 (*C*, C-4), 80.0 (*C*H, C-5), 71.4 (*C*, C-10), 70.4 (*C*H, C-8), 50.5 (*C*, C-7), 43.8 (*C*H, C-2), 36.1 (*C*H_2_, C-3), 35.0 (*C*H_2_, C-9), 22.7 (*C*H_3_, C-14), 22.0 (*C*H_3_, C-15), 20.9 (*C*H_3_, OCOCH_3_-8), and 20.8 (*C*H_3_, OCOCH_3_-5); HRESIMS: *m*/*z* 419.1302 [M+Na]^+^ (calculated for C_19_H_24_O_9_Na, 419.1318, error_ppm_ = −3.8).

Compound **5** was a pale yellow amorphous solid; LC-UV [acetonitrile in 0.1% aqueous formic acid] λ_max_ 218 nm; ^1^H NMR (CDCl_3_, 300 MHz, at 295 K): *δ* 6.46 (1H, s, H-13a), 5.66 (1H, s, H-13b), 4.90 (1H, d, *J* = 5.4 Hz, H-5), 4.84 (1H, dd, *J* = 3.2, 2.8 Hz, H-8), 4.36 (1H, d, *J* = 5.4 Hz, H-6), 3.26 (1H, dd, *J* = 16.8, 2.8 Hz, H-9a), 3.09 (1H, brs, OH-1), 2.80 (1H, d, *J* = 4.2 Hz, H-2), 2.62 (1H, dd, *J* = 12.7, 4.2 Hz, H-3a), 2.17 (3H, s, OCOCH_3_-5), 2.09 (1H, dd, *J* = 16.6, 3.2 Hz, H-9b), 2.08 (3H, s, OCOCH_3_-10), 1.99 (3H, s, OCOCH_3_-8), 1.66 (3H, s, CH_3_-14), 1.63 (1H, d, *J* = 12.7 Hz, H-3b), and 1.32 (3H, s, CH_3_-15); ^13^C NMR (CDCl_3_, 75 MHz, at 295 K): *δ* 169.9 (*C*O, OCOCH_3_-5), 169.6 (*C*O, OCOCH_3_-10), 168.8 (*C*O, OCOCH_3_-8), 167.8 (*C*O, C-12), 136.8 (*C*, C-11), 125.5 (*C*H_2_, C-13), 104.6 (*C*, C-1), 84.0 (*C*H, C-6), 83.3 (*C*, C-4), 81.3 (*C*, C-10), 80.0 (*C*H, C-5), 69.3 (*C*H, C-8), 50.4 (*C*, C-7), 44.5 (*C*H, C-2), 36.2 (*C*H_2_, C-3), 30.4 (*C*H_2_, C-9), 22.7 (*C*H_3_, OCOCH_3_-10), 22.0 (*C*H_3_, C-15), 20.8 (*C*H_3_, OCOCH_3_-5), 20.6 (*C*H_3_, OCOCH_3_-8), and 19.1 (*C*H_3_, C-14); HRESIMS: *m*/*z* 461.1409 [M+Na]^+^ (calculated for C_21_H_26_O_10_Na, 461.1424, error_ppm_ = −3.3).

Compound **6** was a pale yellow amorphous solid; LC-UV [acetonitrile in 0.1% aqueous formic acid] λ_max_ 247 nm; ^1^H NMR (CDCl_3_, 300 MHz, at 295 K): *δ* 6.46 (1H, s, H-13a), 5.66 (1H, s, H-13b), 4.75 (1H, d, *J* = 6.0 Hz, H-5), 4.74 (1H, dd, *J* = 4.2, 2.3 Hz, H-8), 4.32 (1H, d, *J* = 6.0 Hz, H-6), 3.00 (1H, d, *J* = 4.6 Hz, H-2), 2.56 (1H, dd, *J* = 17.0, 2.3 Hz, H-9a), 2.32 (1H, dd, *J* = 12.8, 4.6 Hz, H-3a), 2.16 (3H, s, OCOCH_3_-8), 2.03 (3H, s, OCOCH_3_-5), 1.97 (1H, dd, *J* = 17.0, 4.2 Hz, H-9b), 1.79 (1H, d, *J* = 12.8 Hz, H-3b), 1.52 (3H, s, H-4′), 1.45 (3H, s, H-5′), 1.41 (3H, s, CH_3_-14), and 1.29 (3H, s, CH_3_-15); ^13^C NMR (CDCl_3_, 75 MHz, at 295 K): *δ* 169.8 (*C*O, OCOCH_3_-8), 169.4 (*C*O, OCOCH_3_-5), 168.0 (*C*O, C-12), 136.8 (*C*, C-11), 125.1 (*C*H_2_, C-13), 111.0 (*C*, C-1), 106.1 (*C*, C-2′), 82.3 (*C*, C-4), 81.7 (*C*H, C-6), 79.4 (*C*H, C-5), 76.4 (*C*, C-10), 69.4 (*C*H, C-8), 49.4 (*C*, C-7), 43.2 (*C*H, C-2), 35.5 (*C*H_2_, C-9), 35.4 (*C*H_2_, C-3), 29.4 (*C*H_3_, C-4′), 28.8 (*C*H_3_, C-5′), 25.5 (*C*H_3_, C-14), 21.0 (*C*H_3_, C-15), 20.8 (*C*H_3_, OCOCH_3_-8), and 20.7 (*C*H_3_, OCOCH_3_-5); HRESIMS: *m*/*z* 459.1624 [M+Na]^+^ (calculated for C_22_H_28_O_9_Na, 459.1631, error_ppm_ = −1.5).

For the semisynthesis of 5-hydroxy-hirsutinolide (**7**), reaction condition *c(i)*, BiCl_3_ (323.5 mg, 10.3 mmol, 4.5 molar eq.) was added to a solution of glaucolide B (**1**; 100.1 mg, 0.23 mmol, 1 molar eq.) in anhydrous CH_2_Cl_2_ (2 mL) under an atmosphere of argon (Ar), and the mixture monitored with TLC was stirred for 3 h at room temperature. The reaction mixture was extracted with CH_2_Cl_2_ (3 × 15 mL) via liquid–liquid partition and the organic layer was dried with anhydrous Na_2_SO_4_, filtered through filter paper, and concentrated under reduced pressure, producing compound **7** (80.5 mg, 77.4%). No further purification was necessary.

Compound **7** was a colorless amorphous solid; ^1^H NMR (CDCl_3_, 300 MHz, at 295 K): *δ* 5.90 (1H, dd, *J* = 3.5, 3.4 Hz, H-8), 5.11 (1H, d, *J* = 9.1 Hz, H-6), 4.91 (1H, d, *J* = 12.6 Hz, H-13a), 4.74 (1H, d, *J* = 12.6 Hz, H-13b), 4.30 (1H, brs, OH-1), 3.57 (1H, d, *J* = 9.1 Hz, H-5), 3.27 (1H, brs, OH-5), 2.58 (1H, dd, *J* = 15.6, 3.5 Hz, H-9a), 2.23–2.09 (2H, m, H-3a and H-3b), 2.18–2.08 (1H, m, H-2a), 2.15 (1H, d, *J* = 15.6 Hz, H-9b), 2.09 (3H, s, OCOCH_3_-13), 2.06 (3H, s, OCOCH_3_-8), 2.04 (3H, s, OCOCH_3_-10), 1.88–1.78 (1H, m, H-2b), 1.62 (3H, s, CH_3_-14), and 1.50 (3H, s, CH_3_-15); ^13^C NMR (CDCl_3_, 75 MHz, at 295 K): *δ* 171.7 (*C*O, OCOCH_3_-10), 171.2 (*C*O, C-12), 170.3 (*C*O, OCOCH_3_-13), 169.7 (*C*O, OCOCH_3_-8), 168.6 (*C*, C-7), 123.1 (*C*, C-11), 108.8 (*C*, C-1), 87.0 (*C*, C-10), 86.5 (*C*, C-4), 82.0 (*C*H, C-6), 75.9 (*C*H, C-5), 68.4 (*C*H, C-8), 55.7 (*C*H_2_, C-13), 41.8 (*C*H_2_, C-9), 34.5 (*C*H_2_, C-3), 33.9 (*C*H_2_, C-2), 22.2 (*C*H_3_, OCOCH_3_-10), 21.9 (*C*H_3_, C-15), 20.9 (*C*H_3_, OCOCH_3_-13), 20.6 (*C*H_3_, OCOCH_3_-8), and 17.5 (*C*H_3_, C-14); HRESIMS: *m*/*z* 457.1686 [M+H]^+^ (calculated for C_21_H_29_O_11_, 457.1710, error_ppm_ = −5.2).

For reaction condition *c(ii)*, TFA (10 µL, 0.13 mmol, 1.4 molar eq.) was added to a 1 mL solution of glaucolide B (**1**; 39.7 mg, 0.09 mmol, 1 molar eq.) in anhydrous CH_2_Cl_2_ at −2 °C. The reaction mixture monitored with TLC was stirred for 24 h at room temperature. A saturated NaHCO_3_ aqueous solution (5 mL) was added to the reaction mixture, which was extracted with CHCl_3_ (6 × 5 mL) via liquid–liquid partition, and the organic layer was dried with anhydrous Na_2_SO_4_, filtered through filter paper, and concentrated under reduced pressure. The crude material was purified via silica gel CC using an isocratic eluent composed of *n*-hexane/acetone (70:30, *v*/*v*), producing derivative **7** (15.5 mg, 37.5%), although with a lower reaction yield than *c(i)*.

For the semisynthesis of 1,8-diacetoxy-1(4),7(10)-diepoxy-5-hydroxygermacr-11(13)-en-6(12)-o1ide (**8**), 1N K_2_CO_3_ aqueous solution (725 µL, 0.36 mmol, 5.5 molar eq.) was added to a solution of 5-hydroxy-hirsutinolide (**7**; 30.0 mg, 0.07 mmol, 1 molar eq.) in THF (1.5 mL), and the mixture monitored with TLC was refluxed at 45 °C for 2 h (condition *d*). The reaction mixture was extracted with CH_2_Cl_2_ (3 × 15 mL) via liquid–liquid partition, and the organic layer was dried with anhydrous Na_2_SO_4_, filtered through filter paper, and concentrated under reduced pressure, yielding derivative **8** (8.8 mg, 33.7%). No further purification was necessary.

Compound **8** was a colorless amorphous solid; LC-UV [acetonitrile in 0.1% aqueous formic acid] λ_max_ 246 nm; ^1^H NMR (CDCl_3_, 300 MHz, at 295 K): *δ* 6.54 (1H, s, H-13a), 5.78 (1H, s, H-13b), 5.28 (1H, dd, *J* = 8.5, 7.5 Hz, H-8), 4.68 (1H, d, *J* = 10.3 Hz, H-6), 4.00 (1H, d, *J* = 10.3 Hz, H-5), 3.02 (1H, dd, *J* = 13.9, 8.5 Hz, H-9a), 2.63–2.51 (1H, m, H-2a), 2.63–2.51 (1H, m, H-3a), 2.27–2.17 (1H, m, H-2b), 2.15–1.97 (1H, m, H-3b), 2.04 (3H, s, OCOCH_3_-1), 1.96 (3H, s, OCOCH_3_-8), 1.89 (1H, dd, *J* = 13.9, 7.5 Hz, H-9b), 1.44 (3H, s, CH_3_-15), and 1.38 (3H, s, CH_3_-14); ^13^C NMR (CDCl_3_, 75 MHz, at 295 K): *δ* 171.2 (*C*O, OCOCH_3_-8), 168.2 (*C*O, OCOCH_3_-1), 167.7 (*C*O, C-12), 136.9 (*C*, C-11), 129.4 (*C*H_2_, C-13), 112.1 (*C*, C-1), 89.5 (*C*, C-7), 89.2 (*C*, C-4), 85.7 (*C*, C-10), 84.8 (*C*H, C-6), 82.8 (*C*H, C-8), 73.2 (*C*H, C-5), 40.1 (*C*H_2_, C-9), 36.8 (*C*H_2_, C-3), 31.6 (*C*H_2_, C-2), 22.8 (*C*H_3_, C-14), 22.3 (*C*H_3_, OCOCH_3_-1), 21.1 (*C*H_3_, OCOCH_3_-8), and 19.4 (*C*H_3_, C-15); HRESIMS: *m*/*z* 397.1501 [M+H]^+^ (calculated for C_19_H_25_O_9_, 397.1499, error_ppm_ = 0.5).

For the semisynthesis of 8-acetoxy-vernomargolide (**9**), BiCl_3_ (106.1 mg, 0.34 mmol, 5.5 molar eq.) was added to a solution of 5-hydroxy-hirsutinolide (**7**; 28.0 mg, 0.06 mmol, 1 molar eq.) in anhydrous CH_2_Cl_2_ (2 mL), and the mixture was refluxed at 50 °C for 15 h (condition *e*). The reaction mixture monitored with TLC was extracted with CH_2_Cl_2_ (3 × 15 mL) via liquid–liquid partition, and the organic layer was dried with anhydrous Na_2_SO_4_, filtered through filter paper, and concentrated under reduced pressure, providing derivative **9** (14.6 mg, 67.2%). No further purification was necessary. Compound **9** was a pale yellow amorphous solid; LC-UV [acetonitrile in 0.1% aqueous formic acid] λ_max_ 226 nm; ^1^H NMR (CDCl_3_, 300 MHz, at 295 K): *δ* 6.48 (1H, s, H-13a), 5.87 (1H, s, H-13b), 5.10 (1H, d, *J* = 5.5 Hz, H-8), 4.23 (1H, d, *J* = 4.6 Hz, H-6), 3.58 (1H, d, *J* = 4.6 Hz, H-5), 3.37 (1H, d, *J* = 4.4 Hz, H-2), 2.58 (1H, dd, *J* = 16.3, 5.5 Hz, H-9a), 2.30 (3H, s, CH_3_-14), 2.23 (1H, d, *J* = 16.3 Hz, H-9b), 2.06 (3H, s, OCOCH_3_-8), 1.78 (1H, dd, *J* = 13.4, 4.4 Hz, H-3a), 1.68 (1H, d, *J* = 13.4 Hz, H-3b), and 1.45 (3H, s, CH_3_-15); ^13^C NMR (CDCl_3_, 75 MHz, at 295 K): *δ* 209.4 (*C*O, C-1), 169.6 (*C*O, OCOCH_3_-8), 168.6 (*C*O, C-12), 134.4 (*C*, C-11), 126.7 (*C*H_2_, C-13), 96.5 (*C*, C-10), 87.4 (*C*H, C-6), 84.4 (*C*, C-4), 79.6 (*C*H, C-8), 79.1 (*C*H, C-5), 56.7 (*C*, C-7), 53.4 (*C*H, C-2), 43.2 (*C*H_2_, C-9), 35.9 (*C*H_2_, C-3), 26.5 (*C*H_3_, C-14), 21.2 (*C*H_3_, C-15), and 20.9 (*C*H_3_, OCOCH_3_-8); HRESIMS: *m*/*z* 355.1396 [M+H]^+^ (calculated for C_17_H_23_O_8_, 355.1393, error_ppm_ = 0.8).

For the semisynthesis of 5-acetoxy-hirsutinolide (**7a**), 1,5,8-triacetoxy-1(4),7(10)-diepoxy-germacr-11(13)-en-6(12)-o1ide (**8a**), and 5,8-diacetoxy-vernomargolide (**9a**), one-pot, two-step acetylation reactions were performed via the semisynthetic derivatives **7**, **8**, and **9**. Therefore, as an acetylation general condition, DMAP (0.05 molar eq.) was added to a suspension of Ac_2_O (1–3 molar eq.) and Et_3_N (1.5–3.5 molar eq.) in anhydrous CH_2_Cl_2_ (1 mL), and the mixture was stirred at −10 °C for 30 min. After that, each respective SL semisynthetic derivative (**7**, **8**, or **9**) was added to the mixture, which was stirred for 30 min at room temperature (condition f). Water (3 mL) was added to each reaction mixture, which were monitored with TLC while stirring for 30 min at room temperature. Thereafter, the mixture was extracted with CH_2_Cl_2_ (3 × 15 mL) via liquid–liquid partition. The organic layer was dried with anhydrous Na_2_SO_4_, filtered through filter paper, and concentrated under reduced pressure.

For the acetylation reaction of compound **7** (30.0 mg, 0.07 mmol, 1 molar eq.), DMAP (0.4 mg, 0.003 mmol), Ac_2_O (6 µL, 0.07 mmol, 1 molar eq.), and Et_3_N (14 µL, 0.10 mmol, 1.5 molar eq.) were used. The mixture was monitored with TLC while stirring for 0.5 h at room temperature. After the complete consumption of compound **7**, the obtained crude material was purified via silica gel CC using gradient eluent systems composed of acetone in *n*-hexane (30–50%, *v*/*v*) to obtain the isolation of the C-5 acetylated derivative **7a** (12.4 mg, 37.8%). Compound **7a** was a colorless amorphous solid; LC-UV [acetonitrile in 0.1% aqueous formic acid] λ_max_ 225 nm; ^1^H NMR (CDCl_3_, 300 MHz, at 308 K): *δ* 5.92 (1H, dd, *J* = 3.7, 3.5 Hz, H-8), 5.10 (1H, d, *J* = 9.5 Hz, H-6), 4.96–4.88 (1H, m, H-5), 4.92 (1H, d, *J* = 12.7 Hz, H-13a), 4.74 (1H, d, *J* = 12.7 Hz, H-13b), 2.58 (1H, dd, *J* = 15.5, 3.7 Hz, H-9a), 2.33–2.17 (1H, m, H-2a), 2.14 (3H, s, OCOCH_3–_5), 2.13–1.98 (1H, m, H-3a), 2.09 (3H, s, OCOCH_3_-13), 2.09–2.04 (1H, m, H-9b), 2.05 (3H, s, OCOCH_3_-8), 2.04 (3H, s, OCOCH_3_-10), 1.93–1.82 (1H, m, H-3b), 1.88–1.77 (1H, m, H-2b), 1.62 (3H, s, CH_3_-14), and 1.56 (3H, s, CH_3_-15); ^13^C NMR (CDCl_3_, 75 MHz, at 308 K): *δ* 171.4 (*C*O, OCOCH_3_-10), 170.2 (*C*O, OCOCH_3_-5), 170.2 (*C*O, C-12), 170.1 (*C*O, OCOCH_3_-13), 169.5 (*C*O, OCOCH_3_-8), 166.5 (*C*, C-7), 124.0 (*C*, C-11), 109.5 (*C*, C-1), 86.7 (*C*, C-10), 85.0 (*C*, C-4), 79.4 (*C*H, C-6), 73.4 (*C*H, C-5), 68.3 (*C*H, C-8), 55.7 (*C*H_2_, C-13), 41.2 (*C*H_2_, C-9), 34.3 (*C*H_2_, C-3), 33.6 (*C*H_2_, C-2), 22.4 (*C*H_3_, C-15), 22.1 (*C*H_3_, OCOCH_3_-10), 20.8 (*C*H_3_, OCOCH_3_-13), 20.6 (*C*H_3_, OCOCH_3_-5), 20.5 (*C*H_3_, OCOCH_3_-8), and 17.6 (*C*H_3_, C-14); HRESIMS: *m*/*z* 499.1794 [M+H]^+^ (calculated for C_23_H_31_O_12_, 499.1816, error_ppm_ = −4.4).

Compound **8** (7.7 mg, 0.02 mmol, 1 molar eq.), in turn, was treated with DMAP (0.1 mg, 0.001 mmol), Ac_2_O (2 µL, 0.02 mmol, 1 molar eq.), and Et_3_N (4 µL, 0.03 mmol, 1.5 molar eq.). After completion of the reaction as determined via TLC (0.5 h), the corresponding crude material was purified with preparative TLC using an isocratic eluent system composed of *n*-hexane/acetone (70:30, *v*/*v*; Rf ≅ 0.40), producing the isolated C-5 acetylated derivative **8a** (1.6 mg, 18.8%). Compound **8a** was a colorless amorphous solid; LC-UV [acetonitrile in 0.1% aqueous formic acid] λ_max_ 246 nm; ^1^H NMR (CDCl_3_, 300 MHz, at 295 K): *δ* 6.51 (1H, s, H-13a), 5.79 (1H, s, H-13b), 5.33 (1H, d, *J* = 10.4 Hz, H-5), 5.28 (1H, dd, *J* = 8.3, 6.2 Hz, H-8), 4.67 (1H, d, *J* = 10.4 Hz, H-6), 3.02 (1H, dd, *J* = 14.2, 8.3 Hz, H-9a), 2.76–2.62 (1H, m, H-2a), 2.58–2.46 (1H, m, H-3a), 2.27–2.17 (1H, m, H-2b), 2.11 (3H, s, OCOCH_3_-5), 2.04 (3H, s, OCOCH_3_-1), 1.96 (3H, s, OCOCH_3_-8), 1.94–1.87 (1H, m, H-3b), 1.91 (1H, dd, *J* = 14.2, 6.2 Hz, H-9b), 1.45 (3H, s, CH_3_-15), and 1.41 (3H, s, CH_3_-14); ^13^C NMR (CDCl_3_, 75 MHz, at 295 K): *δ* 170.6 (*C*O, OCOCH_3_-8), 170.0 (*C*O, OCOCH_3_-5), 168.0 (*C*O, OCOCH_3_-1), 167.2 (*C*O, C-12), 135.8 (*C*, C-11), 128.7 (*C*H_2_, C-13), 112.0 (*C*, C-1), 89.7 (*C*, C-7), 87.2 (*C*, C-4), 85.5 (*C*, C-10), 81.8 (*C*H, C-8), 81.4 (*C*H, C-6), 71.9 (*C*H, C-5), 39.6 (*C*H_2_, C-9), 36.5 (*C*H_2_, C-3), 31.2 (*C*H_2_, C-2), 22.6 (*C*H_3_, C-14), 22.0 (*C*H_3_, OCOCH_3_-1), 20.8 (*C*H_3_, OCOCH_3_-8), 20.6 (*C*H_3_, OCOCH_3_-5), and 20.0 (*C*H_3_, C-15); HRESIMS: *m*/*z* 439.1612 [M+H]^+^ (calculated for C_21_H_27_O_10_, 439.1604, error_ppm_ = 1.8).

Compound **9** (12.0 mg, 0.03 mmol, 1 molar eq.) was subjected to acetylation reaction using DMAP (0.2 mg, 0.002 mmol), Ac_2_O (10 µL, 0.10 mmol, 3 molar eq.), and Et_3_N (16 µL, 0.12 mmol, 3.5 molar eq.). The medium was stirred at room temperature and the reaction was stopped within 0.5 h after the complete consumption of the substrate as revealed via TLC. The medium was extracted using an immiscible liquid–liquid separation process with CH_2_Cl_2_ (3 × 15 mL). After the filtration and evaporation of CH_2_Cl_2_, compound **9a** was obtained (8.8 mg, 65.7%). Compound **9a** was a colorless amorphous solid; LC-UV [acetonitrile in 0.1% aqueous formic acid] λ_max_ 246 nm; ^1^H NMR (CDCl_3_, 300 MHz, at 295 K): *δ* 6.50 (1H, s, H-13a), 5.84 (1H, s, H-13b), 5.10 (1H, d, *J* = 5.6 Hz, H-8), 4.90 (1H, d, *J* = 6.2 Hz, H-5), 4.30 (1H, d, *J* = 6.2 Hz, H-6), 3.46 (1H, dd, *J* = 3.5, 1.4 Hz, H-2), 2.65 (1H, dd, *J* = 16.4, 5.6 Hz, H-9a), 2.29 (3H, s, CH_3_-14), 2.19 (3H, s, OCOCH_3_-5), 2.18 (1H, d, *J* = 16.4 Hz, H-9b), 2.04 (3H, s, OCOCH_3_-8), 1.82 (1H, d, *J* = 1.4 Hz, H-3a), 1.81 (1H, d, *J* = 3.5 Hz, H-3b), and 1.30 (3H, s, CH_3_-15); ^13^C NMR (CDCl_3_, 75 MHz, at 295 K): *δ* 208.8 (*C*O, C-1), 169.5 (*C*O, OCOCH_3_-5), 169.3 (*C*O, OCOCH_3_-8), 167.8 (*C*O, C-12), 133.3 (*C*, C-11), 126.3 (*C*H_2_, C-13), 96.6 (*C*, C-10), 83.8 (*C*, C-4), 83.7 (*C*H, C-6), 79.3 (*C*H, C-5), 78.8 (*C*H, C-8), 56.4 (*C*, C-7), 52.0 (*C*H, C-2), 42.5 (*C*H_2_, C-9), 36.2 (*C*H_2_, C-3), 26.2 (*C*H_3_, C-14), 20.6 (*C*H_3_, OCOCH_3_-8), 20.6 (*C*H_3_, C-15), and 20.5 (*C*H_3_, OCOCH_3_-5); HRESIMS: *m*/*z* 397.1501 [M+H]^+^ (calculated for C_19_H_25_O_9_, 397.1499, error_ppm_ = 0.5).

For the semisynthesis of 8,10,13-triacetoxy-1(4)-epoxy-germacra-1(2),5(6),7(11)-trien-6(12)-olide (**7b**), pyridine (8 µL, 0.10 mmol, 1.5 molar eq.) was added to a solution of 5-hydroxy-hirsutinolide (**7**; 30.2 mg, 0.07 mmol, 1 molar eq.) in anhydrous CH_2_Cl_2_ (1.5 mL) under an atmosphere of argon at −10 °C. After that, SOCl_2_ (35 µL, 0.48 mmol, 7.3 molar eq.) diluted in 300 µL of anhydrous CH_2_Cl_2_ was added dropwise to the mixture, which was stirred under the same condition for 30 min. Then, the mixture monitored with TLC was refluxed at 60 °C for 3.5 h (condition *g*). The reaction mixture was concentrated under reduced pressure, and the resulting crude material was purified via SPE using silica gel as a stationary phase and CH_2_Cl_2_ as an eluent, obtaining derivative **7b** (11.1 mg, 40.1%). Compound **7b** was a pale yellow amorphous solid; LC-UV [acetonitrile in 0.1% aqueous formic acid] λ_max_ 279 nm; ^1^ H NMR (CDCl_3_, 300 MHz, at 295 K): *δ* 6.83 (1H, d, *J* = 8.3 Hz, H-8), 5.70 (1H, s, H-5), 5.12 (1H, d, *J* = 12.9 Hz, H-13a), 4.87 (1H, d, *J* = 12.9 Hz, H-13b), 4.82 (1H, dd, *J* = 3.2, 1.3 Hz, H-2), 3.08 (1H, d, *J* = 14.2 Hz, H-9a), 2.78 (1H, dd, *J* = 15.4, 1.3 Hz, H-3a), 2.49 (1H, dd, *J* = 15.4, 3.2 Hz, H-3b), 2.33 (1H, dd, *J* = 14.2, 8.3 Hz, H-9b), 2.09 (3H, s, OCOCH_3_-8), 2.08 (3H, s, OCOCH_3_-13), 2.04 (3H, s, OCOCH_3_-10), 1.81 (3H, s, CH_3_-14), and 1.72 (3H, s, CH_3_-15); ^13^C NMR (CDCl_3_, 75 MHz, at 295 K): *δ* 170.2 (*C*O, OCOCH_3_-13), 169.25 (*C*O, OCOCH_3_-10), 169.20 (*C*O, OCOCH_3_-8), 167.2 (*C*O, C-12), 157.8 (*C*, C-1), 153.6 (*C*, C-7), 144.8 (*C*, C-6), 127.2 (*C*, C-11), 123.2 (*C*H, C-5), 94.1 (*C*H, C-2), 85.8 (*C*, C-4), 77.9 (*C*, C-10), 66.2 (*C*H, C-8), 56.4 (*C*H_2_, C-13), 49.3 (*C*H_2_, C-9), 42.4 (*C*H_2_, C-3), 25.2 (*C*H_3_, C-15), 22.1 (*C*H_3_, OCOCH_3_-10), 21.1 (*C*H_3_, C-14), 20.9 (*C*H_3_, OCOCH_3_-13), and 20.8 (*C*H_3_, OCOCH_3_-8); HRESIMS: *m*/*z* 443.1301 [M+Na]^+^ (calculated for C_21_H_24_O_9_Na, 443.1318, error_ppm_ = −3.8).

For the semisynthesis of 8,10,13-triacetoxy-1(4)-epoxy-6-hydroxy-5-keto-germacr-7(11)-en-6(12)-olide (**7c**), 226.6 mg of a mixture (1:1, m/m) of PCC (113.3 mg, 0.52 mmol, 8 molar eq.) and silica gel was added to a solution of 5-hydroxy-hirsutinolide (**7**; 30.0 mg, 0.07 mmol, 1 molar eq.) in anhydrous CH_2_Cl_2_ (2 mL) under an atmosphere of argon, and the mixture was monitored with TLC. The reaction under a stirring condition was completed within 48 h at room temperature (condition *h*). The reaction mixture was dried under reduced pressure, and the obtained crude material was purified via silica gel CC using an isocratic eluent condition of CH_2_Cl_2_/acetone (80:20, *v*/*v*) to obtain derivative **7c** (20.1 mg, 63.3%). Compound **7c** was a pale yellow amorphous solid; LC-UV [acetonitrile in 0.1% aqueous formic acid] λ_max_ 243 and 299 nm; ^1^ H NMR (CDCl_3_, 300 MHz, at 278 K): *δ* 6.81 (1H, s, OH-6), 6.36 (1H, dd, *J* = 11.7, 6.4 Hz, H-8), 4.84 (1H, d, *J* = 12.7 Hz, H-13a), 4.71 (1H, d, *J* = 12.7 Hz, H-13b), 3.32 (1H, dd, *J* = 14.3, 11.7 Hz, H-9a), 2.69–2.56 (1H, m, H-3a), 2.60 (1H, dd, *J* = 14.3, 6.4 Hz, H-9b), 2.54–2.42 (1H, m, H-2a), 2.12–1.99 (1H, m, H-3b), 2.094 (3H, s, OCOCH_3_-13), 2.090 (3H, s, OCOCH_3_-8), 2.04 (3H, s, OCOCH_3_-10), 2.01–1.86 (1H, m, H-2b), 1.75 (3H, s, CH_3_-15), and 1.61 (3H, s, CH_3_-14); ^13^C NMR (CDCl_3_, 75 MHz, at 278 K): *δ* 200.9 (*C*O, C-5), 173.0 (*C*O, OCOCH_3_-8), 170.0 (*C*O, OCOCH_3_-13), 169.4 (*C*O, OCOCH_3_-10), 167.6 (*C*O, C-12), 162.7 (*C*, C-7), 127.9 (*C*, C-11), 111.1 (*C*, C-1), 107.6 (*C*, C-6), 87.2 (*C*, C-4), 86.5 (*C*, C-10), 68.2 (*C*H, C-8), 55.0 (*C*H_2_, C-13), 36.9 (*C*H_2_, C-3), 35.1 (*C*H_2_, C-9), 34.9 (*C*H_2_, C-2), 28.3 (*C*H_3_, C-15), 21.7 (*C*H_3_, C-14), 21.0 (*C*H_3_, OCOCH_3_-10), 20.8 (*C*H_3_, OCOCH_3_-8), and 20.8 (*C*H_3_, OCOCH_3_-13); HRESIMS: *m*/*z* 471.1504 [M+H]^+^ (calculated for C_21_H_27_O_12_, 471.1503, error_ppm_ = 0.2).

For the semisynthesis of 8,10-diacetoxy-4(7)-epoxy-6-hydroxy-1,5-diketo-germacr-11(13)-en-6(12)-olide (**7c.1**), BiCl_3_ (76.3 mg, 0.24 mmol, 5.5 molar eq.) was added to a solution of compound **7c** (20.0 mg, 0.04 mmol, 1 molar eq.) in anhydrous CH_2_Cl_2_ (2 mL), and the mixture was refluxed at 50 °C and monitored with TLC. The starting material was consumed completely within 15 h (condition *e*). The reaction mixture was extracted with CH_2_Cl_2_ (3 × 15 mL) via liquid–liquid partition. The organic layer was dried with anhydrous Na_2_SO_4_, filtered through filter paper, and the solvent was evaporated under reduced pressure. The crude material was purified with preparative TLC using an isocratic eluent mixture composed of *n*-hexane/acetone (60:40, *v*/*v*; Rf ≅ 0.46), obtaining derivative **7c.1** (2.3 mg, 13.2%). Compound **7c.1** was a colorless amorphous solid; LC-UV [acetonitrile in 0.1% aqueous formic acid] λ_max_ 245 nm; ^1^H NMR (CDCl_3_, 300 MHz, at 295 K): *δ* 6.58 (1H, d, *J* = 0.7 Hz, H-13a), 6.06 (1H, d, *J* = 0.7 Hz, H-13b), 5.28 (1H, d, *J* = 9.5 Hz, H-8), 4.90 (1H, brs, OH-6), 2.63 (1H, dd, *J* = 16.0, 9.5 Hz, H-9a), 2.62–2.49 (1H, m, H-3a), 2.54–2.41 (1H, m, H-2a), 2.26–2.14 (1H, m, H-3b), 2.05 (1H, d, *J* = 16.0 Hz, H-9b), 2.03–1.93 (1H, m, H-2b), 2.02 (3H, s, OCOCH_3_-8), 2.01 (3H, s, OCOCH_3_-10), 1.71 (3H, s, CH_3_-14), and 1.28 (3H, s, CH_3_-15); ^13^C NMR (CDCl_3_, 75 MHz, at 295 K): *δ* 205.6 (*C*O, C-5), 203.9 (*C*O, C-1), 169.5 (*C*O, OCOCH_3_-10), 169.1 (*C*O, OCOCH_3_-8), 166.0 (*C*O, C-12), 136.4 (*C*, C-11), 129.7 (*C*H_2_, C-13), 99.3 (*C*, C-6), 84.5 (*C*, C-10), 83.9 (*C*, C-4), 82.2 (*C*, C-7), 66.9 (*C*H, C-8), 40.7 (*C*H_2_, C-9), 34.3 (*C*H_2_, C-3), 29.5 (*C*H_2_, C-2), 26.2 (*C*H_3_, C-15), 21.1 (*C*H_3_, OCOCH_3_-10), 20.9 (*C*H_3_, OCOCH_3_-8), and 18.3 (*C*H_3_, C-14); HRESIMS: *m*/*z* 411.1279 [M+H]^+^ (calculated for C_19_H_23_O_10_, 411.1291, error_ppm_ = −2.9). (For details on UV, NMR, and HRESIMS of all semisynthetic derivatives, see the Appendix A).

### 3.3. Computational Calculation

The structures of the semisynthetic derivatives were drawn using the software *Marvin 19.1.11.0* [66]. The software *Standardizer JChem 16.1.11.0* [67] was used to canonize the structures. This process used a divide-and-conquer approach and converted an arbitrarily chosen chemical structure into a unique notation, added hydrogens, and cleaned the molecular graph in three dimensions. The structure was split into small fragments that were organized into a tree using connectivity information. Conformers generated for the initial structure (represented by the root node in the tree) were optimized. The tree-building process used a proprietary extended version of the Dreiding force field [68]. Geometric optimizations and conformational searches were performed using *Spartan 16* for *Windows* software [69]. The geometry of the chemical structure of each compound was initially optimized with a Merck molecular force field [70], and a new geometric optimization was then performed based on the PM6 semi-empirical method [71]. A systematic search method was used to analyze conformers and select that with the lowest minimum energy using AM1 and a Monte-Carlo algorithm [72]. After that, minimum energies were selected and optimized based on a vibrational mode calculation using density functional theory (DFT) [73]. DFT calculations were performed using the software *Spartan 16* for *Windows* (Wavefunction, Irvine, CA, USA) [73,74]. Each structure studied was examined at the B3LYP/6-31G* level, and the lowest energy structures were selected for the calculations [73,74]. The global minimum on the potential energy surface was used for the determination of each geometry.

### 3.4. Biological Assay

To evaluate the potential antitumoral activity of the compounds, the MTT [3-(4,5-dimethyl-2-thiazolyl)-2,5-diphenyl-2*H*-tetrazolium bromide] method [75] was used to determine the cell viability of SK-MEL-28 (human melanoma cell line), NCI-H460 (human lung carcinoma cell line), SF295 (human glioblastoma cell line), and HUVEC (human umbilical vein endothelial cells) line. Cell viability analysis of murine macrophage J774A.1 was conducted using a resazurin-based cytotoxicity assay (Alamar blue^®^) [76]. All cell lines were acquired from the BCRJ (Rio de Janeiro Cell Bank, Federal University of Rio de Janeiro, Brazil). The cell lines were cultivated in RPMI-1640 or DMEM medium, supplemented with 1.5 g/L of sodium bicarbonate, 10 mM of HEPES, pH 7.4, 100 U/mL of penicillin G, 100 μg/mL of streptomycin, and 10% fetal calf serum at 37 °C in a humidified atmosphere consisting of 95% air and 5% CO_2_ cultures, and greater than 95% of viable cells in trypan blue exclusion tests were used for the experiments. Therefore, cells were seeded (1 × 10^4^/well) in 96-well plates and incubated with increasing concentrations of the compounds for 24 h. After incubation, the former culture medium was replaced with a fresh culture medium with 5 mg/mL of MTT and with 0.1 mg/mL of resazurin for 2 h at 37 °C. MTT formazan crystals were dissolved in DMSO, and the absorbance of each well was measured at 540 nm, using a micro-well system reader. For the control group of each cell line, cells were incubated without the compounds and considered to have 100% cell viability. MTT formazan crystals were dissolved in DMSO, and absorbance was measured at 540 nm using a microplate reader (ELX800, Biotek, Winooski, VT, USA). The half-maximal cytotoxic concentration (CC_50_) values were calculated from a Hill concentration–response curve. Results were expressed as the mean ± SD of three experiments. The selectivity index (SI) was calculated via the ratio between the CC_50_ of non-tumoral cells and the tumoral cell lines. The higher the SI value is, the more selective the compound is to the tumoral cell line. For the resazurin assay, fluorescence was measured at 1 h intervals using a plate multi-reader (Infinite Multireader m200 Tecan) at 530–590 nm. Nonlinear regression of each fraction versus percentage cell viability was performed to obtain a 10% toxicity concentration (a concentration causing 90% cell viability) using GraphPad Prism 8.0 software (GraphPad Software, San Diego, CA, USA).

The in vitro anti-inflammatory activity was performed in murine macrophage J774A.1 and analyzed after modifications to the protocol previously described by Herath and coworkers [77]. The evaluation consisted of measuring the production of nitrite through an indicator of NO synthesis in supernatants from LPS-activated macrophages. A number of 1 × 10^5^ cells/well were seeded in 96-well plates for 24 h, after which time the culture medium was changed, and the CC_10_ concentration (Appendix A) of each compound was added to the medium containing DMEM supplemented with 10% FBS. After 15 min, macrophages were stimulated with 10 μg/mL of LPS and incubated for 24 h to obtain a proinflammatory M1-phenotype. Subsequently, supernatants were collected and total nitrite nitrate (NOx) was measured by adding 100 μL of Griess’ reagent (1% sulfanilamide and 0.1% *N*-(1-naphthyl) ethylenediamine in 0.5% phosphoric acid) [78]. Absorbance was determined with a microplate reader (ELX800, BioTek, Winooski, VT, USA) at 540 nm. In addition, the proinflammatory cytokine IL-6 was determined using an enzyme-linked immunosorbent assay (ELISA) (BD Bioscience, San Jose, CA, USA) according to the manufacturer’s instructions.

## 4. Conclusions

The use of five different acid and basic catalysts, associated with complementary reaction conditions, allowed the production of fourteen semisynthetic SL-derivatives, including ten new compounds (**4**, **5**, **6**, **7a**, **7c**, **7c.1**, **8**, **8a**, **9**, and **9a**), whose different carbocyclic cores were formed via rearrangement reactions and transannular cyclization. As shown in Figure 2, basic conditions favored the formation of the vernojalcanolide-type (cadinanolides **2** and **3**), and acidic conditions allowed the formation of the hirsutinolide (**7**) and the vernomargolide-type (cadinanolides **4**, **5**, and **6**) from the natural SL glaucolide B (**1**). Furthermore, when subjected to acidic conditions, the hirsutinolide derivative (**7**) allowed the formation of a vernomargolide-type (cadinanolide **9**), and under basic conditions, germacranolide core (**8**) formation was favored.

Regarding antiproliferative activity, hirsutinolides **7a** (CC_50_ = 5.0 µM, SI = 2.5) and **7b** (CC_50_ = 11.2 µM, SI = 2.5) and germacranolide **8a** (CC_50_ = 3.1 µM, SI = 3.0) turned out to be the most active against SK-MEL-28 human melanoma cells. The cytotoxic activity seems to improve as the lipophilicity of these compounds increases compared with the starting material after acetylation reactions. With all compounds showing an anti-inflammatory effect to some degree, compounds **7a** and **7c.1** were the most active with the greatest reduction (of about 77%) in NOx and IL-6 levels in pre-treated M1 macrophages.

## Figures and Tables

**Figure 1 molecules-28-01243-f001:**
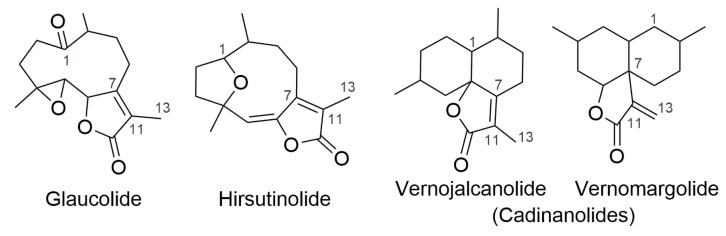
Carbocyclic skeletons of sesquiterpene lactone subtypes.

**Figure 2 molecules-28-01243-f002:**
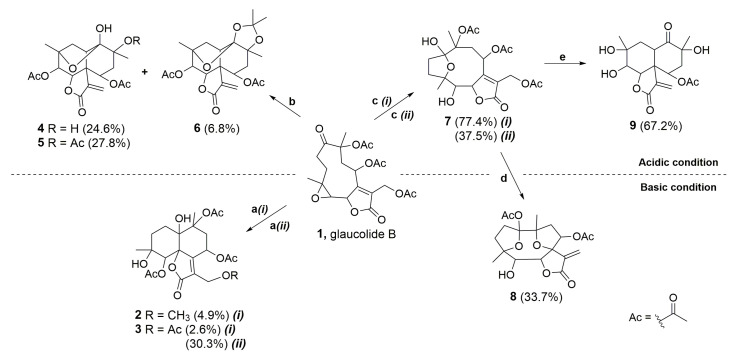
Acidic and basic conditions for the chemical transformation of glaucolide B (**1**). Reaction conditions: (**a**) *(i)* DMAP, MeOH, reflux at 60 °C, 3 h or *(ii)* DMAP, CH_2_Cl_2_, rt, 0.5 h; (**b**) BiCl_3_, Ac_2_O, CH_2_Cl_2_, Ar, rt, 2 h; (**c**) *(i)* BiCl_3_, CH_2_Cl_2_, Ar, rt, 3 h or *(ii)* TFA-H_2_O, CH_2_Cl_2_, −2 °C to rt, 24 h; (**d**) K_2_CO_3_ (aq.), THF, reflux at 45 °C, 2 h; and (**e**) BiCl_3_, CH_2_Cl_2_, reflux at 50 °C, 15 h. Percentage of reaction yield in parentheses.

**Figure 3 molecules-28-01243-f003:**
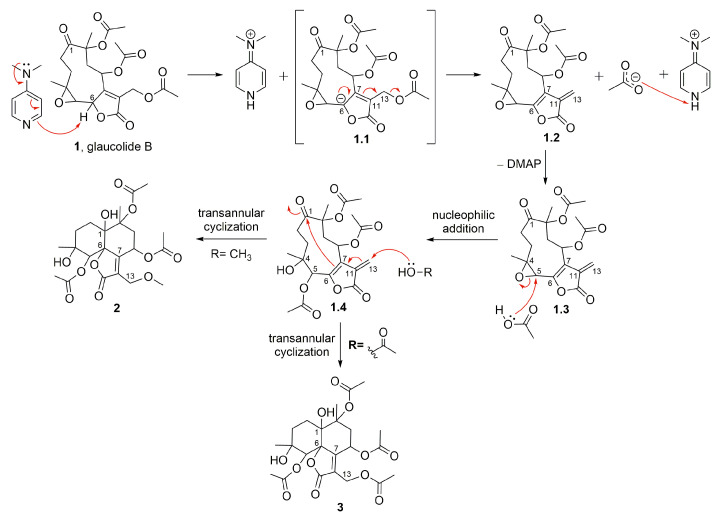
Proposed mechanism for the formation of semisynthetic derivatives **2** and **3** (red arrows illustrate the arrow-pushing mechanism).

**Figure 4 molecules-28-01243-f004:**
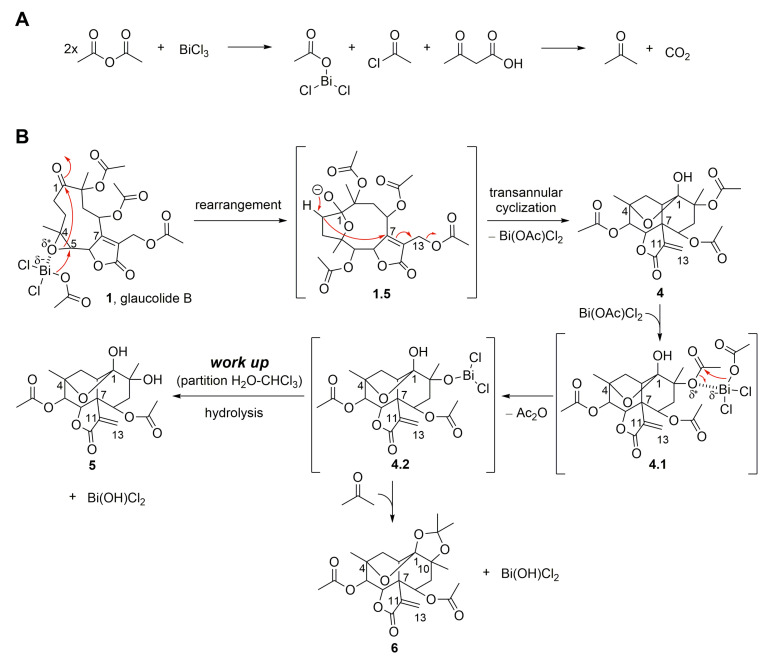
Proposed mechanism for the formation of SL semisynthetic derivatives **4**–**6** involving acetone formation (**A**) and SL-BiCl_3_ complexation (**B**) (red arrows illustrate the arrow-pushing mechanism).

**Figure 5 molecules-28-01243-f005:**
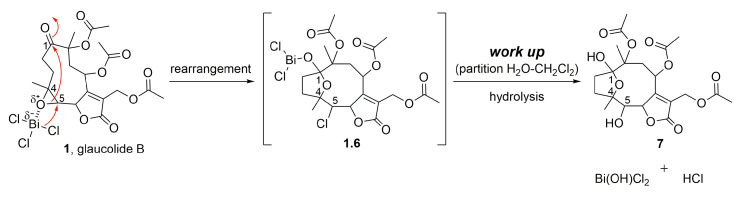
Proposed mechanism for the formation of SL semisynthetic derivative **7** (red arrows illustrate the arrow-pushing mechanism).

**Figure 6 molecules-28-01243-f006:**
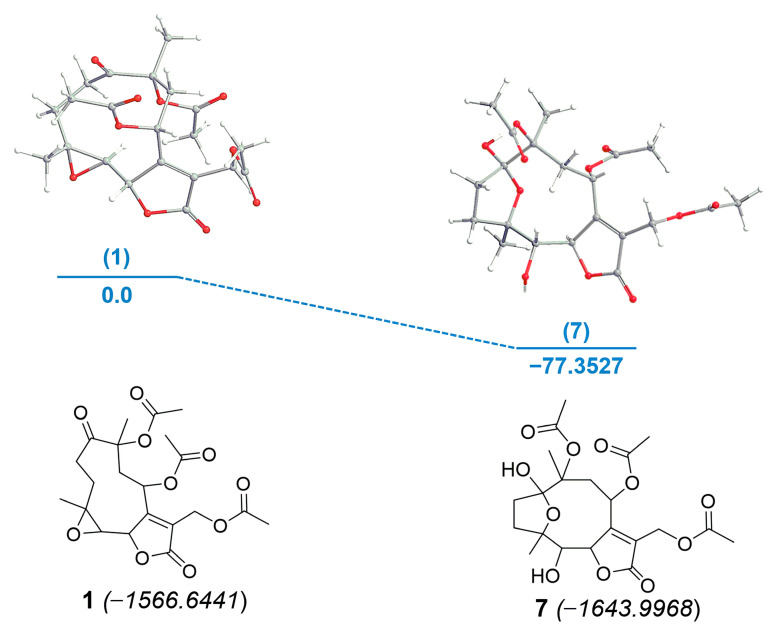
Energy diagram for the formation of SL semisynthetic derivative **7** from **1**. Relative energies [kcal/mol, B3LYP/6-31G*] (blue). Electronic energy values (italic) for compounds **1** and **7** are expressed in kcal/mol. Oxygens are represented as red atoms.

**Figure 7 molecules-28-01243-f007:**
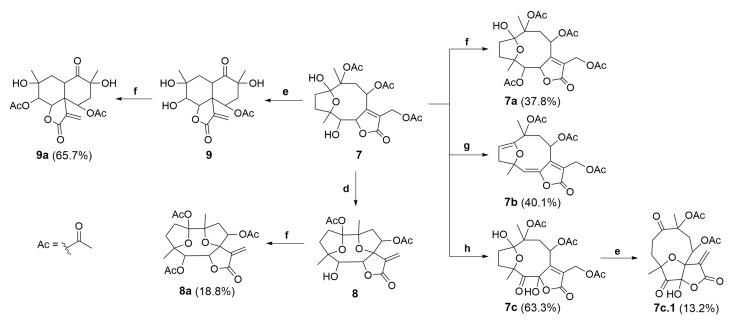
Conditions for the chemical transformations of compounds **7a**–**9a**. Reaction conditions: (**d**) K_2_CO_3_ (aq.), THF, reflux at 45 °C, 2 h; (**e**) BiCl_3_, CH_2_Cl_2_, reflux at 50 °C, 15 h; (**f**) DMAP, Et_3_N, Ac_2_O, CH_2_Cl_2_, −10 °C, 0.5 h; (**g**) SOCl_2_, pyridine, Ar, CH_2_Cl_2_, −10 °C (0.5 h) to reflux at 60 °C (3.5 h); and (**h**) PCC/SiO_2_ (1:1), CH_2_Cl_2_, Ar, rt, 48 h. Percentage of reaction yield in parentheses.

**Figure 8 molecules-28-01243-f008:**
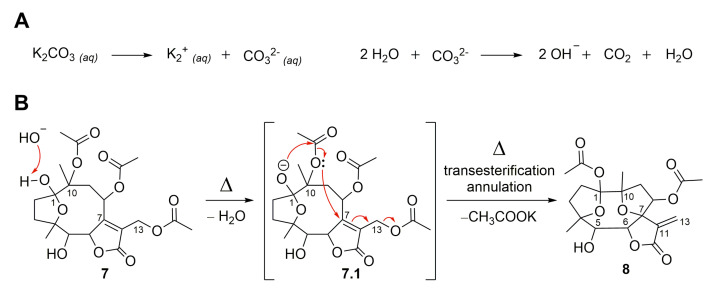
Proposed mechanism for the formation of SL semisynthetic derivative **8** involving an alkaline pH promoted by aqueous K_2_CO_3_ dissociation (**A**) and a rearrangement via transesterification and annulation (**B**) (red arrows illustrate the arrow-pushing mechanism).

**Figure 9 molecules-28-01243-f009:**
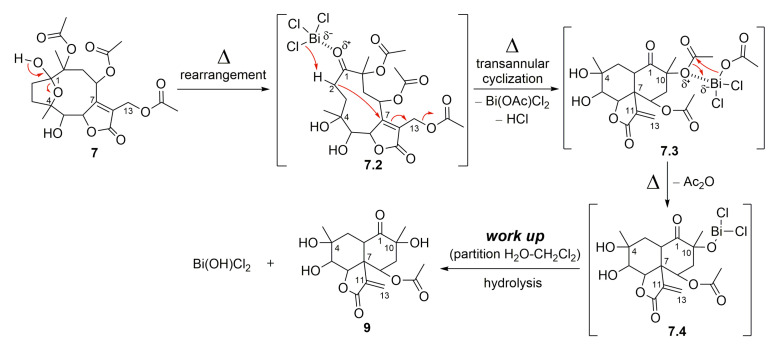
Proposed mechanism for the formation of SL semisynthetic derivative **9** (red arrows illustrate the arrow-pushing mechanism).

**Figure 10 molecules-28-01243-f010:**
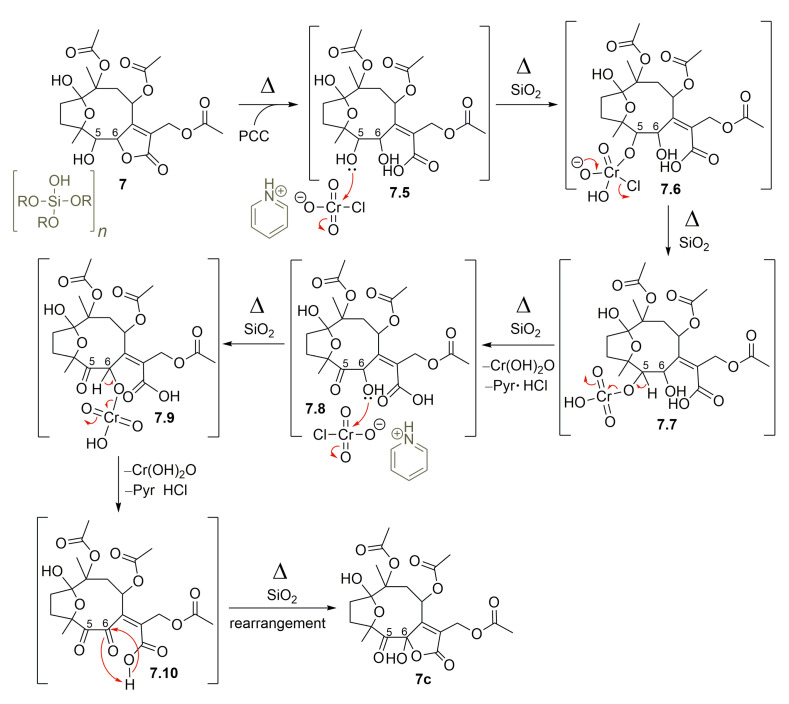
Proposed mechanism for the formation of SL semisynthetic derivative **7c** (red arrows illustrate the arrow-pushing mechanism).

**Figure 11 molecules-28-01243-f011:**
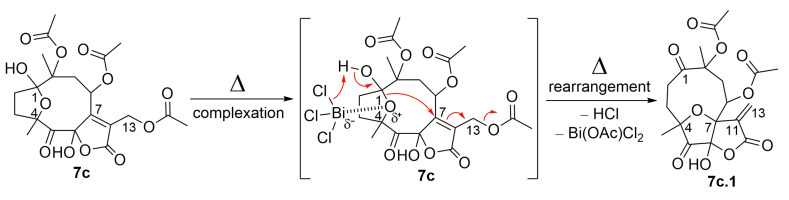
The proposed mechanism for the formation of SL semisynthetic derivative **7c.1** (red arrows illustrate the arrow-pushing mechanism).

**Figure 12 molecules-28-01243-f012:**
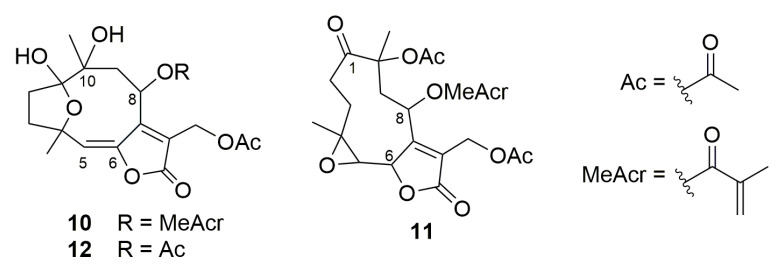
Structures of sesquiterpene lactones **10–12**.

**Figure 13 molecules-28-01243-f013:**
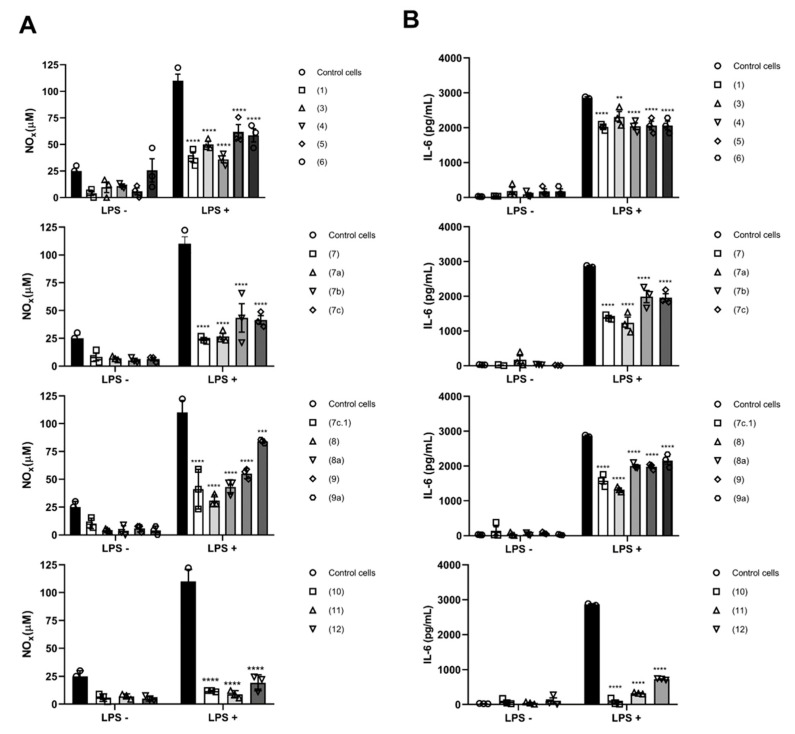
Anti-inflammatory activity evaluated from activated macrophages. (**A**) Total nitrite nitrate (NOx) determination and (**B**) Interleukin 6 (IL-6) in the supernatants of control and LPS-stimulated macrophages (10 μg/mL) in the presence of CC_10_ of compounds **1**, **3**, **4**, **5**, **7**, **7a**, **7b**, **7c**, **7c.1**, **8**, **8a**, **9**, **9a**, **10**, **11**, and **12**. Data expressed as means ± standard deviation of three independent experiments. A two-way analysis of variance followed by Sidak’s multiple comparisons test was performed. ** *p* < 0.01, *** *p* < 0.001 and **** *p* < 0.0001 compared with control group LPS-not stimulated (LPS–) or LPS-stimulated (LPS+). Compound **2** was not evaluated.

**Table 1 molecules-28-01243-t001:** Viability, CC_50_, and selectivity index (SI) values for the cytotoxic activity of the sesquiterpene lactones and semisynthetic derivatives.

Compound	Cell Lines							
	SK-MEL-28			HUVEC	NCI-H460		SF295	
	Viability (%) *^a^*	CC_50_ (μM)	SI	CC_50_ (μM)	Viability (%) *^a^*	CC_50_ (μM)	Viability (%) *^a^*	CC_50_ (μM)
Glaucolide B (**1**)	82.6 ± 4.1	n.d.	-	n.d.	85.1 ± 4.6	n.d.	88.0 ± 4.8	n.d.
Vernojalcanolide derivative (**2**)	87.1 ± 2.6	n.d.	-	n.d.	91.8 ± 3.6	n.d.	86.8 ± 6.3	n.d.
Vernojalcanolide derivative (**3**)	91.1 ± 1.7	n.d.	-	n.d.	88.4 ± 3.4	n.d.	86.1 ± 2.3	n.d.
Vernomargolide derivative (**4**)	96.4 ± 6.9	n.d.	-	n.d.	100.2 ± 2.7	n.d.	94.7 ± 1.1	n.d.
Vernomargolide derivative (**5**)	95.3 ± 4.1	n.d.	-	n.d.	100.3 ± 5.5	n.d.	95.6 ± 3.9	n.d.
Vernomargolide derivative (**6**)	85.9 ± 4.0	n.d.	-	n.d.	90.4 ± 1.5	n.d.	90.4 ± 3.7	n.d.
Hirsutinolide derivative (**7**)	76.2 ± 7.5	n.d.	-	n.d.	89.1 ± 3.9	n.d.	86.7 ± 3.8	n.d.
Hirsutinolide derivative (**7a**)	17.1 ± 3.3	**5.0**	**2.5**	12.7	74.6 ± 1.3	n.d.	86.9 ± 4.3	n.d.
Hirsutinolide derivative (**7b**)	16.0 ± 2.6	11.2	2.5	27.8	73.2 ± 1.0	n.d.	83.3 ± 1.2	n.d.
Hirsutinolide derivative (**7c**)	79.2 ± 6.4	n.d.	-	n.d.	85.7 ± 1.6	n.d.	88.6 ± 4.8	n.d.
Germacranolide derivative (**7c.1**)	77.3 ± 0.3	n.d.	-	n.d.	88.7 ± 6.7	n.d.	87.0 ± 4.7	n.d.
Germacranolide derivative (**8**)	66.9 ± 3.5	n.d.	-	n.d.	87.3 ± 1.6	n.d.	82.0 ± 4.7	n.d.
Germacranolide derivative (**8a**)	5.3 ± 0.3	**3.1**	**3.0**	9.0	53.1 ± 0.0	n.d.	67.3 ± 4.0	n.d.
Vernomargolide derivative (**9**)	74.5 ± 4.9	n.d.	-	n.d.	84.9 ± 0.4	n.d.	85.2 ± 2.8	n.d.
Vernomargolide derivative (**9a**)	73.8 ± 3.1	n.d.	-	n.d.	84.8 ± 0.9	n.d.	85.9 ± 5.0	n.d.
Piptocarphin A (**10**)	23.1 ± 4.2	**6.7**	**1.9**	12.5	104.8 ± 1.6	n.d.	93.2 ± 3.2	n.d.
Glaucolide A (**11**)	56.3 ± 1.4	14.6	1.4	20.4	95.1 ± 4.6	n.d.	91.8 ± 3.3	n.d.
Diacetylpiptocarphol (**12**)	9.0 ± 0.5	**3.8**	**3.3**	12.6	63.8 ± 6.9	n.d.	80.4 ± 3.3	n.d.

*^a^* Results expressed as means ± standard deviation (n = 3); **n.d.**—not determined.

## Data Availability

Not applicable.

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
