# Peer review of "Semisynthetic Sesquiterpene Lactones Generated by the Sensibility of Glaucolide B to Lewis and Brønsted–Lowry Acids and Bases: Cytotoxicity and Anti-Inflammatory Activities"

_molecules, 2023, doi:10.3390/molecules28031243_

Round 1

Reviewer 1 Report

Dear Authors,

The manuscript reports on Semisynthetic sesquiterpene lactones generated by the sensibility of Glaucolide B to Lewis and Brønsted-Lowry acids and bases: Cytotoxicity and anti-inflammatory activities.

The manuscript is well written with adequate data which will be useful to the scientific world. Here are some of my comments to improve the manuscript.

1.  Were the reaction mechanisms studied using spectroscopic methods? Wouldn't it be appropriate to study at least one reaction?

2. Were the NMR spectra of the reaction mixtures measured? If not, what is the guarantee that there was no structural change during the chromatography and that the isolated product is the product that is identical to the product after the reaction?

3. What was the state of the isolated products? For each, it is stated that the residues were obtained. Were they liquids or solids? If they were solids, why aren't the melting points listed?

4. Table 1: SI for 7a is 2.5, not 3.0.

5. The structures of derivatives 10 – 12 are not included in the manuscript. I think it would be appropriate to complete the structures of these derivatives in your manuscript.

6. Biological activity: In my opinion, not only acetyl groups but also the presence of multiple bonds significantly affects the biological activity of the studied substances.

7. It is unusual to report only the biological activity of the studied substances. This activity is usually compared with drugs already known and used in clinical practice. It would be appropriate if the authors could supplement this data.

8. NMR spectra of derivatives 7a, 7b, 8a are not completely pure. In such a case, we can assume that the biological activity of the studied derivatives could have been affected by the impurities present.

9. Was the purity of the synthesized derivatives also determined? How? If not why not? It is necessary to determine the purity of the isolated derivatives that are provided for testing.

10. NMR spectra: It is unusual to report NMR spectra as in this manuscript. NMR chemical shifts are listed from highest to lowest chemical shift. Please correct all NMR spectra in the manuscript.

11. I am a bit confused about the NMR spectra of derivative 2. HSQC peaks for CH2 groups are not visible in the HSQC spectra. It is obvious that the sample was not concentrated enough or the spectra were not measured with enough scans. It is similar in HMBC spectra. These spectra certainly could not be used to determine the structure of derivative 2. I assume that the structure was determined by comparison with the spectra of the other derivatives.

12. As for the presentation of the spectra in SI. It is not appropriate to include the description: "Y axis: 13C projection, in ppm (referenced to TMS - δ 0.00); X-axis: 1H chemical shift, in ppm (referenced to TMS - δ 0.00)", as no 13C projection is shown in the spectra. Moreover, in NMR spectroscopy we do not have X and Y axes, but F1 and F2. This needs to be fixed.

13. Additionally, chemical shifts are given with a decimal point, not a comma.

14. I request that the authors add the 13C spectra of the synthesized and isolated derivatives to the SI.

1

Author Response

Reviewer 1

Dear Authors,

The manuscript reports on Semisynthetic sesquiterpene lactones generated by the sensibility of Glaucolide B to Lewis and Brønsted-Lowry acids and bases: Cytotoxicity and anti-inflammatory activities.

The manuscript is well written with adequate data which will be useful to the scientific world. Here are some of my comments to improve the manuscript.

  1. Were the reaction mechanisms studied using spectroscopic methods? Wouldn't it be appropriate to study at least one reaction?

Response: The proposed mechanisms were conceived based on the substrate and the identified products. We did not use spectroscopic methods to monitor intermediates and the formation of the products. NMR analyses require in this case a most powerful NMR magnet.

Moreover, to perform this study, it will require a new plant collection, extraction, purification of Glaucolide B and semisynthetic reactions.

  1. Were the NMR spectra of the reaction mixtures measured? If not, what is the guarantee that there was no structural change during the chromatography and that the isolated product is the product that is identical to the product after the reaction?

Response: The reaction mixtures were not all monitored by NMR analyses. However, TLC (and some cases also LC-MS analysis) were used to monitor all the reactions. Most importantly, the retention factors revealed that the substrate was completely consumed while new products were being formed.

  1. What was the state of the isolated products? For each, it is stated that the residues were obtained. Were they liquids or solids? If they were solids, why aren’t the melting points listed?

Response: The state has been indicated for each compound in the Materials and Methods section. They were mostly amorphous, that is why their melting points were not provided.

  1. Table 1: SI for 7a is 2.5, not 3.0.

Response: This was corrected in the Table 1 and in the main text.

  1. The structures of derivatives 10 – 12 are not included in the manuscript. I think it would be appropriate to complete the structures of these derivatives in your manuscript.

Response: The structures of 10-12 were provided in the manuscript as “Figure 12”.

  1. Biological activity: In my opinion, not only acetyl groups but also the presence of multiple bonds significantly affects the biological activity of the studied substances.

Response: Justification of the bioactivity was rewritten.

  1. It is unusual to report only the biological activity of the studied substances. This activity is usually compared with drugs already known and used in clinical practice. It would be appropriate if the authors could supplement this data.

Response: Based on our lab protocol, comparison of active with a positive control requires that screened compounds show significant antiproliferative effects (CC50< 1 µM) and high selectivity (SI > 10). As these requirements were not met, we did not find relevant to assess the positive control.

  1. NMR spectra of derivatives 7a, 7b, 8a are not completely pure. In such a case, we can assume that the biological activity of the studied derivatives could have been affected by the impurities present.

Response: 1H NMR spectra of 7a, 7b and 8a contain hexane (from the chromatographic purification) signals at δ 0.88 and 1.26. We strongly believe this residual solvent did not interfere with the bioactivity as we dried all samples in vacuo and kept them in a speed vacuum devise for at 48 hours before submitting to the bio-assays.

  1. Was the purity of the synthesized derivatives also determined? How? If not why not? It is necessary to determine the purity of the isolated derivatives that are provided for testing.

Response: The purity of the synthesized derivatives was monitored by UPLC-UV based on the percentage of each compound peak area in relation to the sum of all areas of the peaks in the chromatogram. We also evaluated the purity based on NMR spectra. As hexane signals were found, all samples were dried using speed vacuum system for 48 h prior bioassays.

  1. NMR spectra: It is unusual to report NMR spectra as in this manuscript. NMR chemical shifts are listed from highest to lowest chemical shift. Please correct all NMR spectra in the manuscript.

Response: The chemical shifts were organized as recommended

  1. I am a bit confused about the NMR spectra of derivative 2. HSQC peaks for CH2 groups are not visible in the HSQC spectra. It is obvious that the sample was not concentrated enough or the spectra were not measured with enough scans. It is similar in HMBC spectra. These spectra certainly could not be used to determine the structure of derivative 2. I assume that the structure was determined by comparison with the spectra of the other derivatives.

Response: Although not visible in the illustrative Figures 2S (HSQC) and 3S (HMBC), weak correlations were observed by increasing the intensity of signals in both 2D correlation maps (because of the very low amount of sample). In addition, since derivative 2 is a substance already described in the literature, its NMR data were compared to those reported in the literature (reference number 38 of the manuscript).

  1. As for the presentation of the spectra in SI. It is not appropriate to include the description: "Y axis: 13C projection, in ppm (referenced to TMS - δ00); X-axis: 1H chemical shift, in ppm (referenced to TMS - δ 0.00)", as no 13C projection is shown in the spectra. Moreover, in NMR spectroscopy we do not have X and Y axes, but F1 and F2. This needs to be fixed.

Response: This was revised in the SI document

  1. Additionally, chemical shifts are given with a decimal point, not a comma.

Response: in decimal numbers, comma was replaced by decimal point

  1. I request that the authors add the 13C spectra of the synthesized and isolated derivatives to the SI.

Response: Technical limitations of our NMR machine (requiring a more powerful NMR magnet and possibly lower temperature analyses) in conjunction to the low amount of sample did not allow to provide 13C NMR spectra with good resolution, Even after 12 hours of NMR analyses, we could have good 13C NMR spectra. Therefore, we found opportune to determine the chemical carbon shifts by using J, J2 and J3 correlations observed in spectra from the two-dimensional experiments (HSQC and HMBC).

Reviewer 2 Report

The research article under title “Semisynthetic sesquiterpene lactones generated by the sensibility of Glaucolide B to Lewis and Brønsted-Lowry acids and bases: Cytotoxicity and anti-inflammatory activities” by de Silva and coworkers investigates the cytotoxicity and anti-inflammatory activities of novel lactones from galucolide. This is a well-written article with plethora of novel compounds chemically characterized and described. The article could be of potential interest to the readers of the Molecules. My recommendation is MAJOR REVISION as there are several points that should be addressed by the authors, as shown below.

The authors should answer the following:

1.     The authors should give a rational explanation on the choice of acids and bases presented in lines 101-102, as these are not the common acids and bases The special emphasis should be put on the effects of different substituents and functionalities that could influence the overall process.

2.     How were the percentages of the products in Figure 2 determined?

3.     The difference in energies in Figure 6 is not the same as in the main text.

4.     What does it mean the “energy proximity”?

5.     Section 2.2. should be removed and integrated into the previous section, as this way it is similar to the Materials and Methods.

6.     Lines 263-270 sound more as something that should be presented in the introduction. This section should only contain the results and discussion for the obtained compounds

7.     The references for the basis set and functional presented in lines 661 and 662 should be given.

8.     More of the quantitative data should be presented in the Conclusion

9.     References should be presented according to the requirements of the journal.

Author Response

Reviewer 2

The research article under title “Semisynthetic sesquiterpene lactones generated by the sensibility of Glaucolide B to Lewis and Brønsted-Lowry acids and bases: Cytotoxicity and anti-inflammatory activities” by de Silva and coworkers investigates the cytotoxicity and anti-inflammatory activities of novel lactones from galucolide. This is a well-written article with plethora of novel compounds chemically characterized and described. The article could be of potential interest to the readers of the Molecules. My recommendation is MAJOR REVISION as there are several points that should be addressed by the authors, as shown below.

The authors should answer the following:

  1. The authors should give a rational explanation on the choice of acids and bases presented in lines 101-102, as these are not the common acids and bases. The special emphasis should be put on the effects of different substituents and functionalities that could influence the overall process.

Response: The choice of acid and bases was discussed/included in the manuscript.

  1. How were the percentages of the products in Figure 2 determined?

Response: In figure 2 the yields of compounds 2-7 were obtained based on the molarity starting compound 1. Those of compounds 8 and 9 were calculated from the molarity of compound 7.

  1. The difference in energies in Figure 6 is not the same as in the main text.

Response: Figure 6 was revised and corrected.

  1. What does it mean the “energy proximity”?

Response: This term was used to infer that the energy difference (difference of enthalpy) observed between the compounds was small.

  1. Section 2.2. should be removed and integrated into the previous section, as this way it is similar to the Materials and Methods.

Response: Section 2.2. was removed and its content was integrated to the previous section (Section 2.1.) as requested. In addition, the following “section 2.3.” was reassigned as a new “Section 2.2.”

  1. Lines 263-270 sound more as something that should be presented in the introduction. This section should only contain the results and discussion for the obtained compounds

Response: This part was moved after the results of the antiproliferative evaluation in the biological part, as a discussion of the important aspects which might be involved as possible mechanism of action for sesquiterpene lactones.

  1. The references for the basis set and functional presented in lines 661 and 662 should be given.

Response: References were provided

  1. More of the quantitative data should be presented in the Conclusion

Response: Quantitative data of biological activity was added for the respective compounds in the Conclusion as requested.

  1. References should be presented according to the requirements of the journal

Response: The references have been reviewed and are in accordance with the Instructions for Authors guide and with very recent articles published in the journal.

Round 2

Reviewer 2 Report

The authors have answered all of the queries by the Reviewers.